# From Continuous Dynamics to Graph Neural Networks: Neural Diffusion and Beyond

**Andi Han**                                                               *andi.han@sydney.edu.au*
*University of Sydney*

**Dai Shi**                                                                *dai.shi@sydney.edu.au*
*University of Sydney*

**Lequan Lin**                                                            *lequan.lin@sydney.edu.au*
*University of Sydney*

**Junbin Gao**                                                           *junbin.gao@sydney.edu.au*
*University of Sydney*

**Reviewed on OpenReview:** *https://openreview.net/forum?id=fPQSxjqa2*

## Abstract

Graph neural networks (GNNs) have demonstrated significant promise in modelling relational data and have been widely applied in various fields of interest. The key mechanism behind GNNs is the so-called message passing where information is being iteratively aggregated to central nodes from their neighbourhood. Such a scheme has been found to be intrinsically linked to a physical process known as heat diffusion, where the propagation of GNNs naturally corresponds to the evolution of heat density. Analogizing the process of message passing to the heat dynamics allows to fundamentally understand the power and pitfalls of GNNs and consequently informs better model design. Recently, there emerges a plethora of works that proposes GNNs inspired from the continuous dynamics formulation, in an attempt to mitigate the known limitations of GNNs, such as oversmoothing and oversquashing. In this survey, we provide the first comprehensive review of studies that leverage the continuous perspective of GNNs. To this end, we introduce foundational ingredients for adapting continuous dynamics to GNNs, along with a general framework for the design of graph neural dynamics. We then review and categorize existing works based on their driven mechanisms and underlying dynamics. We also summarize how the limitations of classic GNNs can be addressed under the continuous framework. We conclude by identifying multiple open research directions.

## 1 Introduction

Graph neural networks (GNNs) (Kipf & Welling, 2017; Veličković et al., 2018; Gilmer et al., 2017) have emerged as one of the most popular choices for processing and analyzing relational data. The main goal of GNNs is to acquire expressive representations at node, edge or graph level, primarily for tasks including but not limited to node classification, link prediction and graph property prediction. Such a learning process requires not only utilizing the node features but also the underlying graph topology. GNNs have been successful in various fields of application where relational structures are critical for learning quality, including recommendation systems (Fan et al., 2019), transportation networks (Jiang & Luo, 2022), particle system (Shlomi et al., 2020), molecule and protein designs (Hoogeboom et al., 2022; Watson et al., 2023), neuroscience (Bessadok et al., 2022) and material science (Xie & Grossman, 2018), among many others.

The effectiveness of GNNs primarily roots in the message passing paradigm, in which information is iteratively aggregated among the neighbouring nodes to update representation of the center node. One prominent

example is the graph convolution network (GCN) (Kipf & Welling, 2017) where each layer updates node representation by taking a degree-weighted average of its neighbours' representation. Although being effective in capturing dependencies among the nodes, the number of layers of message passing requires to be carefully chosen so as to avoid performance degradation. This is unlike the classic (feedforward) neural networks where increasing depth generally leads to improved predictive performance. In particular, due to the nature of message passing, deeper GNNs have a tendency to over-smooth the features, leading to un-informative node representations (Rusch et al., 2023a). On the other hand, shallow GNNs are likely to suffer from information bottleneck where signals from distant nodes (with regards to graph topology) exert little influence on the centered node (Topping et al., 2022; Di Giovanni et al., 2023).

In order to analyze and address the aforementioned issues, many recent works have cast GNNs as discretization of certain continuous dynamical systems, by analogizing propagation through the layers to time evolution of dynamics. Indeed, the idea of viewing neural networks in the continuous limits is not new and has been explored for classic neural networks (E, 2017; Haber & Ruthotto, 2017; Liu & Theodorou, 2019; Li et al., 2022; Ruthotto & Haber, 2020). The continuous formulation provides a unified framework for understanding and designing the propagation of neural networks, leveraging various mathematical tools from control theory and differential equations. One seminal work (Chen et al., 2018) proposes neural ordinary differential equation (Neural ODE), which can be viewed as a continuous-depth residual network (He et al., 2016). Another work explores the connections between the convolutional networks with partial differential equations (PDEs) (Ruthotto & Haber, 2020).

In GNNs, in particular, the message passing mechanism can be viewed as realization of a class of PDEs, namely the heat diffusion equation (Chamberlain et al., 2021a). Such a novel perspective allows to better dissect the behaviours of GNNs. As an example, because heat diffusion is known to dissipate energy and converge to a steady state of heat equilibrium, the phenomenon of oversmoothing corresponds to the equilibrium state of diffusion where heat distributes uniformly across spatial locations. Apart from offering theoretical insights into the evolution of GNNs, the continuous dynamics perspective also allows easy integration of structural constraints and desired properties into the dynamical system, such as energy conservation, (ir)reversibility, and boundary conditions. Meanwhile advanced numerical integrators, like high-order and implicit schemes, can be employed to enhance the efficiency and stability of discretization (Butcher, 2016). Finally, the variety of continuous dynamics grounded with physical substance can inform better designs of GNNs to enhance the representation power and overcome the performance limitations (Chamberlain et al., 2021a; Thorpe et al., 2022; Eliasof et al., 2021; Rusch et al., 2022; Maskey et al., 2023; Zhao et al., 2023b; Choi et al., 2023; Eliasof et al., 2023; Gruber et al., 2023; Kang et al., 2023).

The theory of differential equations and dynamical systems is well-developed with a long-standing history (Verhulst, 2006; Perko, 2013), while graph neural network is a comparatively nascent field, garnering attention within the past decade. This offers great potential in harnessing the rich foundations of the dynamical system theory for enhancing the understanding and functionality of GNNs. In this work, we provide a comprehensive review of existing developments in continuous dynamics inspired GNNs, which we collectively refer to as graph neural dynamics. To the best of our knowledge, this is the first survey on the connections between continuous dynamical systems and graph neural networks. Nevertheless, we would like to highlight a related line of research that utilizes deep learning to solve differential equations (Huang et al., 2022; Gupta et al., 2023; Kumar & Yadav, 2023; Kovachki et al., 2023). In particular, graph neural operators (Anandkumar et al., 2020; Li et al., 2020) employ GNNs to solve PDEs by discretizing the domains and constructing graph structure based on spatial proximity. In contrast, the idealization that we focus in this work is the reverse where PDEs inform the designs of graph neural networks.

**Organization.**   We organize the rest of the paper as follows. In Section 2, we introduce diffusion equations from first principles and then review discrete operators on graphs such as gradient and divergence, which are crucial for formulations of continuous GNNs. This section concludes with the connection between diffusion equation with the famous graph convolutional network. Section 3 then presents the framework for designing GNNs from continuous dynamics and summarizes the existing works based on the underlying physical processes. In Section 4, we explain how the various dynamics help to tackle the shortcomings of existing GNNs, in terms of oversmoothing, oversquashing, poor performance on heterophilic graphs (where

neighbouring nodes do not share similar information), as well as adversarial robustness and training stability. We then in Section 5 discuss the various numerical schemes for propagating and learning the continuous GNN dynamics. In Section 6, we summarize and compare the computational complexities of the dynamics discussed in this work and section 7 discusses empirical benchmarks for evaluating graph neural dynamics. We then conclude the survey by outlining several open research questions and challenges in Section 8.

## 2   From diffusion equations to graph neural networks

A fundamental building block of graph neural networks is message passing where information flows between neighbouring nodes. Message passing has natural connection to diffusion equations, which describe how certain quantities of interest, such as mass or heat disperse spatially, as a function of time.

**Diffusion equation in continuous domains.**   The two laws governing any diffusion equation are *Fick's law* and *mass conservation law*. The former states the diffusion happens from the regions of higher concentration to regions of lower concentration and the rate of diffusion is proportional to the mass difference (the gradient). The latter states the mass cannot be created nor destroyed through the process of diffusion. Formally, let $x : \Omega \times \mathbb{R} \to \mathbb{R}$ represent the mass distribution, i.e., $x(u, t)$ over the spatial locations $u \in \Omega$ and time $t \in \mathbb{R}$. Denote $\nabla x \coloneqq \frac{\partial x}{\partial u}$ as the gradient of mass across the space. Fick's law states that a flux $J$ (quantifying the direction and magnitude of mass movement) points towards lower mass concentrated regions, i.e., $J = -D\nabla x$, where $D$ is the diffusivity coefficient that potentially depends on both space and time. When $\Omega \subseteq \mathbb{R}^d$, we can write $\nabla x = [\frac{\partial x}{\partial u_1}, ... \frac{\partial x}{\partial u_d}]$ and $D \in \mathbb{R}^{d \times d}$ and $J$ corresponds to a flux in $d$ directions. The mass conservation law then leads to the following continuity equation $\frac{\partial x}{\partial t} = -\text{div}J$, where div is the divergence operator that computes the mass changes at certain location and time. Combining the two equations yields the fundamental (heat) diffusion equation

$$\frac{\partial x}{\partial t} = \text{div}(D\nabla x). \tag{1}$$

The diffusion process is called *homogeneous* when the diffusivity coefficient $D$ is space independent, and is called *isotropic* when $D$ depends only on the location, and called *anisotropic* when $D$ depends on both the location and the direction of the gradient. In the case of homogeneous diffusion, we can write the diffusion equation in terms of the Laplacian operator $\Delta \coloneqq \text{div} \cdot \nabla$, i.e., $\frac{\partial x}{\partial t} = D\Delta x$.

**Graphs and discrete differential operators.**   In order to properly characterize diffusion dynamics over graphs, it is necessary to generalize the concepts from differential geometry to the discrete space, such as gradient and divergence.

A graph can be represented by a tuple $\mathcal{G} = (\mathcal{V}, \mathcal{E})$ where $\mathcal{V}, \mathcal{E}$ denote the set of nodes and edges respectively. In this work, we focus on the undirected graphs, i.e., if $(i, j) \in \mathcal{E}$, then $(j, i) \in \mathcal{E}$. Leveraging tools from differential geometry, let $L^2(\mathcal{V})$ and $L^2(\mathcal{E})$ be Hilbert spaces for real-valued functions on $\mathcal{V}$ and $\mathcal{E}$ respectively with the inner product given by

$$\langle f, g \rangle_{L^2(\mathcal{V})} = \sum_{i \in \mathcal{V}} f_i g_i, \qquad \langle F, G \rangle_{L^2(\mathcal{E})} = \sum_{(i,j) \in \mathcal{E}} F_{ij} G_{ij}$$

for $f, g : \mathcal{V} \to \mathbb{R}$ and $F, G : \mathcal{E} \to \mathbb{R}$. This allows to generalize the definitions of gradient and divergence to graph domains (Bronstein et al., 2017). Formally, graph gradient is defined as $\nabla : L^2(\mathcal{V}) \to L^2(\mathcal{E})$ such that $(\nabla f)_{ij} = f_j - f_i$. Here we assume the edges are anti-symmetric, namely $(\nabla f)_{ij} = -(\nabla f)_{ji}$. Graph divergence $\text{div} : L^2(\mathcal{E}) \to L^2(\mathcal{V})$, is defined as the converse of graph gradient such that $(\text{div}F)_i = \sum_{j:(i,j) \in \mathcal{E}} F_{ij}$.

By the discrete nature, graphs and functions/signals on a graph can be compactly represented via vectors and matrices. In a graph with $n$ nodes ($|\mathcal{V}| = n$), each $f \in L^2(\mathcal{V})$ can be written as $n$-dimensional vector $\mathbf{f}$ and $F \in L^2(\mathcal{E})$ can be written as a matrix $\mathbf{F}$ of size $n \times n$ (with nonzero $i, j$-th entry $\mathbf{F}_{i,j}$ only when $(i, j) \in \mathcal{E}$) An adjacency matrix is $\mathbf{A} \in \{0, 1\}^{n \times n}$ such that $\mathbf{A}_{i,j} = 1$ if $(i, j) \in \mathcal{E}$ and 0 otherwise. The degree matrix $\mathbf{D}$ is a diagonal degree matrix with $i$-th diagonal entry $\mathbf{D}_{i,i} = \deg_i = \sum_j \mathbf{A}_{i,j}$. Graph Laplacian is then defined

Table 1: List of commonly used notations

| Notations | Explanations |
|---|---|
| $\mathcal{G} = (\mathcal{V}, \mathcal{E})$ | Graph $\mathcal{G}$ with node set $\mathcal{V}$ and edge set $\mathcal{E}$ |
| $L^2(\mathcal{V}), L^2(\mathcal{E})$ | Hilbert space for functions on $\mathcal{V}, \mathcal{E}$ |
| $\mathbf{X} \in \mathbb{R}^{N \times c}$ | Node signal matrix |
| $\mathbf{A}, \widehat{\mathbf{A}} \in \mathbb{R}^{N \times N}$ | Graph adjacency, normalized adjacency matrix |
| $\widehat{\mathbf{L}} \in \mathbb{R}^{N \times N}$ | Normalized Laplacian matrix |
| $\mathbf{A}(\mathbf{X}) \in \mathbb{R}^{N \times N}$ | Graph attention matrix computed on $\mathbf{X}$ |
| $\deg_i$ | Degree of node $i$ |
| $\mathrm{div}, \nabla$ | Divergence and gradient operator |
| $\mathcal{E}_{\mathrm{dir}}$ | Dirichlet energy. |
| $\frac{\partial \mathbf{X}}{\partial t}, \frac{\partial^2 \mathbf{X}}{\partial t^2}$ | First-order and second-order time derivative of node signal |
| $\odot, \cdot, \otimes$ | elementwise product/ elementwise product (broadcast across channels)/ tensor product |

as $\mathbf{L} = \mathbf{D} - \mathbf{A}$ and one can verify

$$\mathrm{div}(\nabla \mathbf{f}) = \sum_{j:(i,j) \in \mathcal{E}} (f_j - f_i) = -\mathbf{L}\mathbf{f}.$$

Further, graph gradient can be represented with the incidence matrix $\mathbf{G} \in \mathbb{R}^{e \times n}$. Specifically $\mathbf{G}_{k,i} = 1$ if edge $k$ enters node $i$, $-1$ if edge $k$ leaves node $i$ and 0 otherwise. Graph divergence is thus given by $-\mathbf{G}^\top$, which is the negative adjoint of the gradient. The edge direction in $\mathbf{G}$ can be arbitrarily chosen for undirected graphs because the Laplacian is indifferent to the choice of direction as $\mathbf{L} = \mathbf{G}^\top \mathbf{G}$.

**Graph diffusion and graph neural networks.** In this work, we consider $\mathbf{x} : \mathcal{V} \to \mathbb{R}^c$ as a multi-channel signal (or feature) over the node set. We denote $\mathbf{x}_i \in \mathbb{R}^c$ as the signal over node $i$ and let $\mathbf{X} \in \mathbb{R}^{n \times c}$ collects the signals over all the nodes. The previously defined discrete gradient and divergence operators lead to the following graph diffusion process, which can be seen as a discrete version of heat diffusion equation in (1):

$$\frac{\partial \mathbf{x}_i}{\partial t} = \mathrm{div}(D \nabla \mathbf{X})_i = \sum_{j:(i,j) \in \mathcal{E}} D(\mathbf{x}_i, \mathbf{x}_j, t)(\mathbf{x}_j - \mathbf{x}_i),$$

where the diffusivity $D(\mathbf{x}_i, \mathbf{x}_j, t)$ is often scalar-valued and positive, which applies channel-wise. In the special case of homogeneous diffusion, i.e., $D(\mathbf{x}_i, \mathbf{x}_j, t) = 1$, the diffusion process can be written as $\frac{\partial \mathbf{x}_i}{\partial t} = \mathrm{div}(\nabla \mathbf{X})_i = -(\mathbf{L}\mathbf{X})_i$. This process is known as the Laplacian smoothing where the signals become progressively smooth by within-neighbourhood averaging. The solution to the diffusion equation is given by the heat kernel, i.e., $\mathbf{X}(t) = \exp(-t\mathbf{L})\mathbf{X}(0)$. In fact, the graph heat equation can also be derived as the gradient flow of the so-called Dirichlet energy, which measures the local variations: $\mathcal{E}_{\mathrm{dir}}(\mathbf{X}) = \sum_{(i,j) \in \mathcal{E}} \|\mathbf{x}_i - \mathbf{x}_j\|^2 = \mathrm{tr}(\mathbf{X}^\top \mathbf{L}\mathbf{X})$.

The graph diffusion process is related to the famous graph convolutional network (GCN) (Kipf & Welling, 2017), where the latter can be viewed as taking the Euler discretization of the former with a unit stepsize. In addition, GCN defines diffusion with (symmetrically) normalized adjacency $\widehat{\mathbf{A}} = \mathbf{D}^{-1/2}\mathbf{A}\mathbf{D}^{-1/2}$ and Laplacian $\widehat{\mathbf{L}} = \mathbf{I} - \widehat{\mathbf{A}}$. The use of normalized Laplacian $\widehat{\mathbf{L}}$ in place of the combinatorial Laplacian $\mathbf{L}$ switches the dynamics from being homogeneous to isotropic (where diffusion is weighted by node degrees). More precisely, the discretized dynamics gives the update $\mathbf{X}^{\ell+1} = \mathbf{X}^\ell - \widehat{\mathbf{L}}\mathbf{X}^\ell = \widehat{\mathbf{A}}\mathbf{X}^\ell$. It can be readily verified that GCN corresponds to the gradient flow of a normalized Dirichlet energy $\mathcal{E}_{\mathrm{dir}}(\mathbf{X}) = \sum_{(i,j) \in \mathcal{E}} \|\mathbf{x}_i/\sqrt{\deg_i} - \mathbf{x}_j/\sqrt{\deg_j}\|^2 = \mathrm{tr}(\mathbf{X}^\top \widehat{\mathbf{L}}\mathbf{X})$. Additional channel mixing $\mathbf{W}$ and nonlinear activation $\sigma(\cdot)$ are added, leading to a single GCN layer, $\mathbf{X}^{\ell+1} = \sigma(\widehat{\mathbf{A}}\mathbf{X}^\ell \mathbf{W}^\ell)$.

In the more general setup with anisotropic diffusion coefficients $D(\mathbf{x}_i, \mathbf{x}_j, t)$, the diffusion process relates to the general form of message passing neural network (MPNN) (Gilmer et al., 2017). Taking the Euler

discretization of the diffusion equation with stepsize $\tau$, we can rewrite the process as

$$\mathbf{x}_i^{\ell+1} = \mathbf{x}_i^\ell + \tau \sum_{j:(i,j)\in\mathcal{E}} D(\mathbf{x}_i^\ell, \mathbf{x}_j^\ell)(\mathbf{x}_j^\ell - \mathbf{x}_i^\ell) = U^\ell\big(\mathbf{x}_i^\ell, \sum_{j:(i,j)\in\mathcal{E}} M^\ell(\mathbf{x}_i^\ell, \mathbf{x}_j^\ell)\big),$$

where $M(\mathbf{x}_i, \mathbf{x}_j) = D(\mathbf{x}_i, \mathbf{x}_j)(\mathbf{x}_j - \mathbf{x}_i)$ represents the message passing between two nodes $i, j$ and $U(\mathbf{x}_i, \mathbf{m}_i) = \mathbf{x}_i + \tau\mathbf{m}_i$ represents the message aggregation within the neighbourhood of node $i$. The channel-mixing matrix and nonlinear activation can be added for both the message passing and aggregation steps.

We finally remark that there exist GNNs that cannot be easily interpreted within the continuous diffusion framework. This includes MPNNs where the update involves operations such as concatenation (Kearnes et al., 2016), gated recurrent unit (Li et al., 2015). In addition, spectral GNNs, such as ChebGCN (Defferrard et al., 2016), also do not have a natural correspondence in the physical process. Hence we exclude such GNNs from the discussion of this work.

## 3 A general framework of continuous dynamics informed GNNs beyond diffusion

Diffusion dynamics has been shown to underpin the design of message passing and graph convolutional networks. While showing promise in many applications, vanilla diffusion dynamics may suffer from performance degradation due to the following four types of causes:

- *Oversmoothing*: signals become increasingly similar as the propagation of GNN.

- *Oversquashing*: message cannot be easily communicated

- *Heterophily*: neighbouring nodes do not share similar features

- *Instability*: Sensitivity to adversarial perturbation and vanishing or exploding gradients when increasing depth.

This has motivated the consideration of alternative dynamics other than isotropic diffusion, including anisotropic diffusion, diffusion with source term, geometric diffusion, oscillation, convection, advection and reaction. Many of them are directly adapted from the existing physical processes (as we discuss in this work).

This section summarizes existing developments on graph neural dynamics under a general framework and categories the literature by the underlying continuous system. We follow the same notations in Section 2 where a graph is represented as $\mathcal{G} = (\mathcal{V}, \mathcal{E})$ with $|\mathcal{V}| = n$ and $|\mathcal{E}| = e$. The graph signals or features are encoded in a matrix $\mathbf{X}$, which is in general time-dependent. Unless mentioned otherwise, we omit the time dependence and treat $\mathbf{X}$ as $\mathbf{X}(t)$ for notation clarity. We also denote $\mathbf{X}^0 = \mathbf{X}(0)$ as the initial conditions for the system.

The general framework for designing continuous graph dynamics is given as follows, which relates the time derivatives of the signals with spatial derivatives on graphs.

$$\left[\frac{\partial \mathbf{X}}{\partial t}, \frac{\partial^2 \mathbf{X}}{\partial t^2}\right] = \mathcal{F}_\mathcal{G}(\mathbf{X}, \nabla \mathbf{X}), \tag{2}$$

where $\mathcal{F}_\mathcal{G}$ is a spatial coupling function that is usually parameterized by neural networks. The initial condition of the system is generally the input graph signals (after an encoder). We have summarized and compared existing works discussed in the survey in Table 2 in terms of the driving mechanisms and problems addressed. As we shall see, most of the existing works consider the first-order time derivative while some works explore the second-order time derivative to encode oscillatory systems.

As a summary, *anisotropic diffusion* provides more flexibility in adapting to different graph types by allowing different diffusivity coefficients, which may depend on the current state of graph signals. This can be achieved by explicitly factoring out the edge indicators or implicitly modelled via attention mechanism. The main driving mechanism is the diffusion with learnable coefficients, i.e., $\frac{\partial \mathbf{X}}{\partial t} = \text{div}\big(\mathbf{A}(\mathbf{X}) \cdot \nabla \mathbf{X}\big)$, for some learnable diffusion coefficients. *Oscillatory processes* model the acceleration through second-order differential equations,

i.e., $\frac{\partial^2 \mathbf{X}}{\partial t^2} = F_{\mathcal{G}}(\mathbf{X}, \nabla \mathbf{X})$. The energy conservation of such processes help to avoid converging to trivial solutions of constants and thus circumvent oversmoothing. This also prevents energy from exploding, which helps stabilize the training. *Non-local dynamics* allows information exchange at a longer distance, and thus have a wider receptive field at each step. This is in contrast to diffusion based dynamics where communications happen locally. This can be beneficial for tasks where long-range dependencies are crucial, such as node classification for heterophilic graphs. *External forces* can be further injected to modulate the direction, magnitude and even the nature of information flow on a graph, which leads to $\frac{\partial \mathbf{X}}{\partial t} = F_{\mathrm{diff}}(\mathbf{X}) + F_{\mathrm{ext}}(\mathbf{X})$, for some diffusion dynamics $F_{\mathrm{diff}}$ and some mechanisms governed by external forces $F_{\mathrm{ext}}$. When equipping graphs with additional geometric structures, the resulting *geometry-underpinned dynamics* offers a brand-new perspective in controlling the behaviours of GNNs by modifying the underlying geometries. That is, $\frac{\partial \mathbf{X}}{\partial t} = F_{\mathcal{G},\mathcal{M}}(\mathbf{X})$ where the dynamics rely on the additional geometry $\mathcal{M}$ imposed on the graph. Furthermore, dynamics can often be interpreted as inherently minimizing some energy functional and thus can be seen as its *gradient flow*. This suggests directly modifying the energy rather than the dynamics to achieve desired properties. At last, *multi-scale diffusion* separates the diffusion dynamics for low-pass and high-pass filters, and thus achieve greater control for learning different frequency components.

## 3.1 Anisotropic diffusion

The first class of dynamics, *anisotropic diffusion*, generalizes the isotropic diffusion in GCN, offering great flexibility in controlling the local diffusivity patterns. In image processing, anisotropic diffusion has been extensively applied for low-level tasks, such as image denoising, restoration and segmentation (Weickert et al., 1998). In particular, the Perona-Malik model (Perona & Malik, 1990) sets the diffusivity coefficient $D \propto |\nabla x|^{-1}$, which is often called the edge indicator as it preserves the sharpness of signals by slowing down diffusion in regions of high variations.

The idea of using anisotropic diffusion to define continuous GNN dynamics has been firstly considered by Poli et al. (2019) and formalized by Chamberlain et al. (2021a), where a class of graph neural diffusion (GRAND) dynamics is proposed. The anisotropic diffusion process is formally given by

$$\mathrm{GRAND}: \quad \frac{\partial \mathbf{X}}{\partial t} = \mathrm{div}\big(\mathbf{A}(\mathbf{X}) \cdot \nabla \mathbf{X}\big),$$

where $\mathbf{A}(\mathbf{X}) \in \mathbb{R}^{n \times n}$ encodes the anisotropic diffusivity along the edges. Here $\nabla \mathbf{X} \in \mathbb{R}^{n \times n \times c}$ collects the gradient along edges and we use $\cdot$ to represent the elementwise multiplication of diffusivity coefficients broadcast across the channel dimension. The coefficient is determined by the feature similarity, i.e., $\mathbf{A}(\mathbf{X}) = [a(\mathbf{x}_i, \mathbf{x}_j)]_{(i,j) \in \mathcal{E}}$ where $a(\mathbf{x}_i, \mathbf{x}_j) = (\mathbf{W}_K \mathbf{x}_i)^\top \mathbf{W}_Q \mathbf{x}_j$ computes the dot product attention with learnable parameters $\mathbf{W}_K, \mathbf{W}_Q$. Softmax normalization is performed on $\mathbf{A}(\mathbf{X})$ to ensure it is right-stochastic (row sums up to one). Notably, the explicit-Euler discretization of GRAND corresponds to the propagation of graph attention network (GAT) (Veličković et al., 2018). Several versions of GRAND are proposed to render the dynamics more adaptable compared to the discretized version in (Veličković et al., 2018). In particular, the diffusivity matrix $\mathbf{A}(\mathbf{X})$ can be fixed as $\mathbf{A}(\mathbf{X}^0)$ that only depends on the initial features. This leads to a GAT propagation with shared attention weights. In addition, $\mathbf{A}(\mathbf{X})$ can vary according to a dynamically rewired edge set based on the attention score, i.e., $\mathcal{E} \leftarrow \{(i,j) : (i,j) \in \mathcal{E}, a(\mathbf{x}_i, \mathbf{x}_j) > \rho\}$ for some threshold $\rho > 0$. Instructively, this interprets the graph structure as discrete realization of certain underlying domains where graph rewiring dynamically changes the spatial discretization.

Building on the formulation of GRAND, BLEND (Chamberlain et al., 2021b) further augments the input signals $\mathbf{x}_i$ with position coordinates $\mathbf{u}_i$ for each node. The diffusion process then becomes

$$\mathrm{BLEND}: \quad \frac{\partial [\mathbf{X}, \mathbf{U}]}{\partial t} = \mathrm{div}(\mathbf{A}([\mathbf{X}, \mathbf{U}]) \cdot \nabla [\mathbf{X}, \mathbf{U}]).$$

The joint diffusion over both positions and signals is motivated by diffusion on Riemannian manifolds with the Laplace-Beltrami operator, in which the gradient, diffusivity and divergence all depend on the Riemannian metric (that varies according to the position). In a similar vein, augmenting the position information while diffusing on graphs produces joint evolution of features as well as topology, which further allows graph rewiring to improve the information flow. In particular, based on the evolved positional information, the

Table 2: Summary of continuous dynamics informed graph neural networks, including the driving mechanism and the problems addressed, including oversmoothing (OSM), oversquashing (OSQ), graph heterophily (HETERO), and stability (STAB) with respect to perturbation or with training, such as gradient vanishing or explosion associated with increased depth. Note we show the problems addressed either theoretically or empirically in the paper (unless proven otherwise in subsequent literature). More detailed discussions are in Section 4.

| | Methods | Mechanism | Problems Tackled | | | |
| --- | --- | --- | --- | --- | --- | --- |
| | | | OSM | OSQ | HETERO | STAB |
| Anisotropic diffusion | GRAND (Chamberlain et al., 2021a) | Attention diffusivity & rewiring | | ✔* | | ✔ |
| | BLEND (Chamberlain et al., 2021b) | Position augmentation | | ✔* | | ✔ |
| | Mean Curvature (Song et al., 2022) Beltrami (Song et al., 2022) | Non-smooth edge indicators | | | | ✔ |
| | $p$-Laplacian (Fu et al., 2022) | $p$-Laplacian regularization | ✔† | | | |
| | DIFFormer (Wu et al., 2023b) | Full graph transformer | | ✔ | | |
| | DIGNN (Fu et al., 2023) | Parameterized Laplacian | ✔† | | | |
| | GRAND++ (Thorpe et al., 2022) | Source term | ✔ | | | |
| | PDE-GCN$_D$ (Eliasof et al., 2021) | Nonlinear diffusion | ✔ | | ✔ | |
| | GIND (Chen et al., 2022) | Implicit nonlinear diffusion | | | ✔ | ✔ |
| Oscillation | PDE-GCN$_H$ (Eliasof et al., 2021) | Wave equation | ✔ | | ✔ | |
| | GraphCON (Rusch et al., 2022) | Damped coupled oscillation | ✔ | | ✔ | ✔ |
| Non-local dynamics | FLODE (Maskey et al., 2023) | Fractional Laplacian | ✔ | ✔ | ✔ | |
| | QDC (Markovich, 2023) | Quantum diffusion | | ✔ | ✔ | |
| | TIDE (Behmanesh et al., 2023) | Learnable heat kernel | | ✔ | | |
| | G2TN (Toth et al., 2022) | Hypo-elliptic diffusion | | ✔ | | |
| Diffusion with external forces | CDE (Zhao et al., 2023b) | Convection diffusion | | | ✔ | |
| | GREAD (Choi et al., 2023) | Reaction diffusion | ✔ | | ✔ | |
| | ACMP (Wang et al., 2023) | Allen-Chan retraction with negative diffusivity | ✔ | | ✔ | |
| | ODNet (Lv et al., 2023) | Diffusion with confidence | ✔ | | ✔ | |
| | ADR-GNN (Eliasof et al., 2023) | Advection reaction diffusion | ✔ | | ✔ | |
| | G$^2$ (Rusch et al., 2023b) | Diffusion gating | ✔ | | ✔ | |
| | MHKG (Shao et al., 2023) | Reverse diffusion | ✔ | ✔ | ✔ | |
| | A-DGN (Gravina et al., 2023) | Anti-symmetric weight | ✔ | | ✔ | ✔ |
| Geometry underpinned dynamics | NSD (Bodnar et al., 2022) | Sheaf diffusion | ✔ | | ✔ | |
| | Hamiltonian$_G$, etc. (Gruber et al., 2023) | Bracket dynamics with higher-order cliques | ✔ | | | |
| | HamGNN (Kang et al., 2023) (Zhao et al., 2023a) | Learnable Hamiltonian dynamics | ✔ | | | ✔ |
| Gradient flow | GRAFF (Giovanni et al., 2023) | Parameterized gradient flow | ✔ | | ✔ | |
| Multi-scale diffusion | GradFUFG (Han et al., 2022) | Separated diffusion for low-pass and high-pass at different scales | ✔ | | ✔ | |

*The ability of the methods to mitigate oversquashing is through graph rewiring. †$p$-Laplacian and DIGNN avoids oversmoothing provided the input dependent regularization (i.e., a soure term) is added.

graph is dynamically rewired as $\mathcal{E} \leftarrow \{(i,j) : d_\mathcal{C}(\mathbf{u}_i, \mathbf{u}_j) < r\}$ or by k-nearest neighbour graph. The positional information can be pre-computed by personalized PageRank (Gasteiger et al., 2019), deepwalk (Perozzi et al., 2014) or even learned hyperbolic embeddings (Chami et al., 2019).

In BLEND, the diffusivity is given by the attention score over both the positional features and signals, which is in fact the core idea of transformers (Vaswani et al., 2017) that augments samples with positional encoding. Wu et al. (2023b) develop a transformer-based diffusion process (called DIFFormer) where attention diffusivity coefficients are computed on a fully connected graph $\mathcal{V} \times \mathcal{V}$. The input graph (represented via input adjacency matrix $\mathbf{A}^0$) serves as a geometric prior that augments the learned attention matrix.

$$\text{DIFFormer}: \quad \frac{\partial \mathbf{X}}{\partial t} = \text{div}_{\mathcal{V} \times \mathcal{V}}\big((\mathbf{A}^0 + \mathbf{A}(\mathbf{X})) \cdot \nabla \mathbf{X}\big),$$

where $\text{div}_{\mathcal{V}\times\mathcal{V}}(\mathbf{F})_i = \sum_{j\in\mathcal{V}} \mathbf{F}_{i,j}$ denotes the divergence operator on the complete graph and $\mathbf{A}(\mathbf{X})$ computes the diffusivity by a transformer block (Vaswani et al., 2017), different to the ones used by GRAND and BLEND. A recent work (Wu et al., 2023a) has shown the evolution via both local message passing through $\mathbf{A}^0$ and global attention diffusion through $\mathbf{A}(\mathbf{X})$ improves the generalization under topological distribution shift, i.e., when training and test graph topology differs.

Apart from its connection to transformer-based dynamics, BLEND is in fact motivated by a geometric evolution known as Beltrami flow where the diffusion over a non-Euclidean domains also depends on the varying metric. In (Song et al., 2022), the Beltrami flow, along with mean-curvature flow (a geometric flow where the movement is governed by the mean curvature at each point on the surface), is generalized to graph domains by explicitly factorizing out the edge indicator $\|\delta\mathbf{x}_i\| := \sqrt{\sum_{j:(i,j)\in\mathcal{E}} \|\mathbf{x}_j - \mathbf{x}_i\|^2}$. Formally, let $\mathbf{S}_{\text{mc}}(\mathbf{X}), \mathbf{S}_{\text{bel}}(\mathbf{X})$ be the diffusivity coefficients of mean curvature and Beltrami diffusion, with their elements defined as $[\mathbf{S}_{\text{mc}}(\mathbf{X})]_{i,j} = \frac{1}{\|\delta\mathbf{x}_i\|} + \frac{1}{\|\delta\mathbf{x}_j\|}$ and $[\mathbf{S}_{\text{bel}}(\mathbf{X})]_{i,j} = \frac{1}{\|\delta\mathbf{x}_i\|^2} + \frac{1}{\|\delta\mathbf{x}_i\|\|\delta\mathbf{x}_j\|}$. The diffusion processes Song et al. (2022) propose are

$$\text{Mean Curvature}: \quad \frac{\partial\mathbf{X}}{\partial t} = \text{div}\Big(\big(\mathbf{A}(\mathbf{X})\odot\mathbf{S}_{\text{mc}}(\mathbf{X})\big)\cdot\nabla\mathbf{X}\Big),$$

$$\text{Beltrami}: \quad \frac{\partial\mathbf{X}}{\partial t} = \text{div}\Big(\big(\mathbf{A}(\mathbf{X})\odot\mathbf{S}_{\text{bel}}(\mathbf{X})\big)\cdot\nabla\mathbf{X}\Big),$$

where $\mathbf{A}(\mathbf{X})$ is computed from the dot product attention following the previous works (Chamberlain et al., 2021a;b). For both dynamics, non-smooth signals are preserved by slowing down diffusion where signal abruptly changes. As commented by the paper, positional information can be added in a similar way as BLEND (Chamberlain et al., 2021b). It should be noticed that although BLEND originates from the Beltrami flow, the dynamics turns out to be the same as GRAND, augmented with positional embeddings. In contrast, Song et al. (2022) explicitly separates the edge indicator out from the attention matrix $\mathbf{A}(\mathbf{X})$.

A more general $p$-Laplacian based graph neural network is proposed by Fu et al. (2022) where the diffusion is derived from a $p$-Laplacian regularization framework.

$$p\text{-Laplacian}: \quad \frac{\partial\mathbf{X}}{\partial t} = \text{div}\big(\|\nabla\mathbf{X}\|^{p-2}\cdot\nabla\mathbf{X}\big) - \mu(\mathbf{X} - \mathbf{S}),$$

with $\mathbf{S}$ as a source term and $\mu > 0$ controlling the regularization strength. The diffusivity $\|\nabla\mathbf{X}\|^{p-2}$ is an $n\times n$ matrix with elements $[\|\nabla\mathbf{X}\|^{p-2}]_{i,j} = \|\mathbf{x}_j - \mathbf{x}_i\|^{p-2}$ if $(i,j)\in\mathcal{E}$. The injection of source information can be understood physically as the heat exchange from the system to the outside. In the paper the source term is simply the input feature matrix, i.e., $\mathbf{S} = \mathbf{X}^0$. When $p = 2$, the diffusion reduces to the heat diffusion with classic Laplacian. When $p = 1$, the dynamics recovers the mean curvature flow (although the definition slightly differs from the one in (Song et al., 2022)). As a result, with properly selected $p$, the process is flexible in adapting to different types of graphs and able to perverse the boundaries without oversmoothing the signals. It is worth mentioning that unlike previous works that use discretization to update $\mathbf{X}$, the paper directly solves for the equilibrium state by setting $\frac{\partial\mathbf{X}}{\partial t} = 0$, which leads to an implicit graph diffusion layer given by $p$-Laplacian message passing.

The $p$-Laplacian diffusion corresponds to the gradient flow of a $p$-Dirichlet energy, defined by $\mathcal{E}^p_{\text{dir}}(\mathbf{X}) = \sum_{(i,j)\in\mathcal{E}} \|\mathbf{x}_i - \mathbf{x}_j\|^p$ given with a regularization term $\|\mathbf{X} - \mathbf{S}\|^2$. A similar idea has been considered in (Dan et al., 2023) where the Dirichlet energy is replaced with the total variation on graph gradients, which leverages the $L_1$ norm (which is different to the case of $p = 1$ in the $p$-Laplacian diffusion (Song et al., 2022)). A dual-optimization scheme is introduced due to the non-differentiablity of the objective at zero.

A recent paper (Fu et al., 2023) parameterizes the graph gradient and divergence instead of the diffusivity coefficients as in previous works, and defines a parameterized graph Laplacian for diffusion process. In particular, Fu et al. (2023) consider weighted inner products for both the $L^2(\mathcal{V})$ and $L^2(\mathcal{E})$, i.e., $\langle f, g\rangle_{L^2(\mathcal{V})} = \sum_{i\in\mathcal{V}} \chi_i f_i g_i$ and $\langle F, G\rangle_{L^2(\mathcal{E})} = \phi_{i,j} F_{i,j} G_{i,j}$. The graph gradient is defined as $(\nabla_\Theta f)_{i,j} := \psi_{i,j}(f_j - f_i)$, which also leads to a notion of graph divergence $(\text{div}_\Theta F)_i = \frac{1}{2\chi_i}\sum_{j\in\mathcal{N}_i} \psi_{i,j}\phi_{i,j}(F_{i,j} - F_{j,i})$, where $\chi_i, \phi_{i,j}, \psi_{i,j}$ are strictly positive real-valued functions on nodes and edges respectively. Here we denote $\nabla_\Theta, \text{div}_\Theta$ to emphasize that the gradient and divergence operators are parameterized. Because the graph gradient is parameterized

and may not be anti-symmetric, i.e., $(\nabla_\Theta f)_{i,j} \neq -(\nabla_\Theta f)_{j,i}$, the divergence encodes directional information. The paper also parameterizes the weighting functions to be node-dependent, i.e., $\chi_i = \chi_i(\mathbf{x}_i), \phi_{i,j} = \phi_{i,j}(\mathbf{x}_i, \mathbf{x}_j)$ and $\psi_{i,j} = \psi_{i,j}(\mathbf{x}_i, \mathbf{x}_j)$, which involves learnable parameters. The diffusion process the paper considers is thus

$$\text{DIGNN}: \quad \frac{\partial \mathbf{X}}{\partial t} = \text{div}_\Theta(\nabla_\Theta \mathbf{X}) - \mu(\mathbf{X} - \mathbf{S}),$$

where a regularization term $\|\mathbf{X} - \mathbf{S}\|^2$ is added similarly as in (Fu et al., 2022).

The idea of adding an energy source has also been considered in GRAND++ (Thorpe et al., 2022) based on the framework of anisotropic diffusion of GRAND:

$$\text{GRAND++}: \quad \frac{\partial \mathbf{X}}{\partial t} = \text{div}(\mathbf{A}(\mathbf{X}) \cdot \nabla \mathbf{X}) + \mathbf{S}$$

where $\mathbf{S} \in \mathbb{R}^{n \times c}$ is a source term. The paper proposes a random walk viewpoint to show that, without the source term, the dynamics reduces to GRAND and is guaranteed to converge to a stationary distribution independent of the initial conditions $\mathbf{X}^0$. Each row of the source term $\mathbf{S}$ is defined as $\mathbf{s}_i = \sum_{j \in \mathcal{I}} \delta_{ij}(\mathbf{x}_j^0 - \bar{\mathbf{x}}^0)$ with $\mathcal{I} \subseteq \mathcal{V}$ a selected node subset used as the source term and $\bar{\mathbf{x}}^0 = \frac{1}{|\mathcal{I}|} \sum_{j \in \mathcal{I}} \mathbf{x}_j^0$ is the average signal. $\delta_{ij}$ denotes the initial transition probability from node $j$ to $i$. By construction, the limiting signal distribution is close to an interpolation of the source signals in the selected subset $\mathcal{I}$. Similar idea of source term injection has also been considered in earlier work (Xhonneux et al., 2020), which can be seen as the homogeneous diffusion with both a source term and a residual term. The proposed model (called CGNN) follows the dynamics $\frac{\partial \mathbf{X}}{\partial t} = -\mathbf{L}\mathbf{X} + \mathbf{X}\mathbf{W} + \mathbf{X}^0$, for some learnable channel mixing matrix $\mathbf{W}$.

Anisotropic diffusion is generally nonlinear in the sense that diffusivity depends nonlinearly on the mass along the evolution. This is the case in edge-preserving dynamics as the diffusivity explicitly depends nonlinearly on the gradient. GRAND-based dynamics is also nonlinear as long as the attention coefficients is computed for each timestep. In (Eliasof et al., 2021; Chen et al., 2022), additional nonlinearity is further incorporated by factoring the diffusivity $D$ as the composition of a linear operator $\mathcal{K}$ and its adjoint $\mathcal{K}^*$, i.e., $D = \mathcal{K}^*\mathcal{K}$. Then a pointwise nonlinearity function $\sigma(\cdot)$ is added, leading to

$$\frac{\partial \mathbf{X}}{\partial t} = \text{div}\big(\mathcal{K}^*\sigma(\mathcal{K}\nabla \mathbf{X})\big).$$

In the case when $\sigma$ is the identity map, the dynamics recovers the anisotropic diffusion. Such a nonlinear system has been firstly proposed by Ruthotto & Haber (2020) for defining convolutional residual networks for images. In (Eliasof et al., 2021), the idea is generalized to graphs by defining $\mathcal{K}$ as learnable pointwise convolution. Specifically, the dynamics, called PDE-GCN$_\text{D}$, can be written in terms of the gradient and divergence operator as follows.

$$\text{PDE-GCN}_\text{D}: \quad \frac{\partial \mathbf{X}}{\partial t} = -\mathbf{G}^\top \mathbf{K}^\top \sigma(\mathbf{K}\mathbf{G}\mathbf{X}),$$

where $\mathbf{K}$ is a learnable parameter and $\mathbf{G}$ is the gradient operator defined in Section 2. It can be readily observed that when $\sigma$ is identity and $\mathbf{K} = \mathbf{I}$, the dynamics reduces to the heat diffusion implemented by GCN.

In the follow-up work (Chen et al., 2022), the linear operator $\mathcal{K}$ (parameterized by $\mathbf{K}$) is applied over the channel space instead of the edge space, i.e.,

$$\text{GIND}: \quad \frac{\partial \mathbf{X}}{\partial t} = -\mathbf{G}^\top \sigma(\mathbf{G}\mathbf{X}\mathbf{K}^\top)\mathbf{K}.$$

Motivated by Gu et al. (2020), Chen et al. (2022) consider an implicit propagation of GIND as $\mathbf{Z} = -\mathbf{G}^\top \sigma\big(\mathbf{G}(\mathbf{Z} + b_\mathbf{\Omega}(\mathbf{X}^0))\mathbf{K}^\top\big)\mathbf{K}$, where $b_\mathbf{\Omega}(\cdot)$ is an affine transformation with parameter $\mathbf{\Omega}$. The model corresponds to a refinement process for the flux $\mathbf{Z}$ and the output is given by a decoder over $\mathbf{Z} + \mathbf{X}^0$. It has been shown the equilibrium state of the implicit diffusion corresponds to the minimizer of a convex objective function provided the nonlinearity is monotone and Lipschitz and $\mathbf{K} \otimes \mathbf{G}$ is upper bounded in norm. This result guarantees the convergence of the dynamics and allows structural constraints to be embedded to the dynamics by explicitly modifying the objective.

### 3.2 Oscillations

The phenomenon of oscillation has been widely found in physics, which primarily features the repetitive motion. Unlike diffusion that dissipates energy, oscillatory system often preserves energy and is thus reversible. Oscillatory processes are often modelled as second-order ordinary/partial differential equations. One simple example of oscillatory process is characterized by the wave equation $\frac{\partial^2 x}{\partial t^2} = c\Delta x$, which is a hyperbolic PDE. The wave equation has been considered in (Eliasof et al., 2021) for defining dynamics on graphs, which follows the nonlinear formalism in (Ruthotto & Haber, 2020):

$$\text{PDE-GCN}_{\text{H}}: \quad \frac{\partial^2 \mathbf{X}}{\partial t^2} = \text{div}\big(\mathcal{K}^* \sigma(\mathcal{K}\nabla\mathbf{X})\big) = -\mathbf{G}^\top \mathbf{K}^\top \sigma(\mathbf{KGX}).$$

In addition, GraphCON (Rusch et al., 2022) considers more general oscillatory dynamics which combines a damped oscillating process with a coupling function. That is,

$$\text{GraphCON}: \quad \frac{\partial^2 \mathbf{X}}{\partial t^2} = \sigma(F_{\mathcal{G}}(\mathbf{X})) - \gamma\mathbf{X} - \alpha\frac{\partial\mathbf{X}}{\partial t},$$

where $F_{\mathcal{G}}(\mathbf{X})_i = F_{\mathcal{G}}(\mathbf{x}_i, \{\mathbf{x}_j\}_{j\in\mathcal{N}_i})$ is the coupling function, $\alpha \geq 0$ and $\sigma(\cdot)$ is some nonlinear activation. When $\sigma(F_{\mathcal{G}}(\mathbf{X})) = 0$, $\alpha = 0$, the system reduces to the classic harmonic oscillation for each node independently. Adding the damping term $-\frac{\partial\mathbf{X}}{\partial t}$ mimics the frictional force that diminishes the oscillation. Finally, due to the presence of interdependence between the nodes, the coupling function is required to model the interactions between the nodes. The paper mainly considers two choices of coupling function, with isotropic and anisotropic diffusion (which leads to GCN and GAT respectively). Formally, $F_{\mathcal{G}}(\mathbf{X})_i = \sum_{j:(i,j)\in\mathcal{E}} \mathbf{A}(\mathbf{X})_{i,j}\mathbf{x}_j$ represents the message passing with normalized adjacency or learned attention scores. When $\gamma = 1$, $\sigma$ is identity, the GraphCON can be rewritten as $\frac{\partial^2\mathbf{X}}{\partial t} = \text{div}(\mathbf{A}(\mathbf{X})\cdot\nabla\mathbf{X}) - \alpha\frac{\partial\mathbf{X}}{\partial t}$, which is effectively the wave equation with a damping term. GraphCON is flexible in that the coupling term can accommodate arbitrary message passing scheme. In addition, it possesses greater expressive power by showing the GNN induced by the coupling function approaches the steady state of GraphCON, while the latter explores the entire trajectory.

### 3.3 Non-local dynamics

We have so far focused on local dynamics, in the sense that the diffusion or oscillation happens locally within the neighbourhood. Thus it usually requires sufficiently large timestep for one node's influence to reach a distant node (with respect to graph topology). Graph rewiring employed in GRAND and BLEND can be utilized to enable long-range diffusion by modifying the graph topology. This section explores various dynamics-based formulations that transcend the local community when propagating information, resulting in non-localized dynamics.

**Fractional Laplacian.** Fractional Laplacian (Pozrikidis, 2018; Lischke et al., 2020) has been effective to represent complex anomalous processes, such as fluids dynamics in porous medium (Caffarelli & Vazquez, 2011). A recent work (Maskey et al., 2023) utilizes the fractional graph Laplacian/adjacency matrix $-\widehat{\mathbf{A}}^\alpha$ (for some $\alpha \in \mathbb{R}$) in order to define a non-local diffusion process as

$$\text{FLODE}: \quad \frac{\partial\mathbf{X}}{\partial t} = -\widehat{\mathbf{A}}^\alpha\mathbf{X}.$$

where $\widehat{\mathbf{A}}$ is the symmetric normalized adjacency matrix. A critical difference compared to $p$-Laplacian in terms of the order is that here $\alpha$ can be fractional, instead of being restricted to integers. The fractional Laplacian is often dense, and thus the corresponding diffusion is non-local where long-range interactions are captured. When coupled with a symmetric channel mixing matrix $\mathbf{W}$, i.e., $\frac{\partial\mathbf{X}}{\partial t} = -\widehat{\mathbf{A}}^\alpha\mathbf{XW}$, the flexibility in the choice of $\alpha$ allows dynamics to accommodate both smoothing and sharpening effects, which avoids oversmoothing and is suited for heterophilic graphs. The work also extends the formulation to directed graphs and correspondingly defines the notions of oversmoothing and Dirichlet energy with the asymmetric Laplacian. On directed graphs, the fractional Laplacian is defined through singular value decomposition, i.e.,

$\widehat{\mathbf{A}}^\alpha := \mathbf{U}\Sigma^\alpha\mathbf{V}^H$, where $\mathbf{U}, \mathbf{V} \in \mathbb{C}^{n \times n}$ are unitary singular vectors and $\Sigma \in \mathbb{R}^{n \times n}$ contains singular values on the diagonal. Similar idea of using SVD for directed graph processing has been explored in (Zou et al., 2023). Finally, the paper also considers a Schrödinger equation based diffusion as $\frac{\partial \mathbf{X}}{\partial t} = i\widehat{\mathbf{A}}^\alpha\mathbf{X}$, where $i = \sqrt{-1}$ represents the imaginary unit.

**Quantum diffusion kernel.** The Schrödinger's equation $i\frac{\partial \psi(u,t)}{\partial t} = \mathcal{H}\psi(u,t)$ has also been considered in (Markovich, 2023), where $\mathcal{H}$ is the Hamiltonian operator (composed of a kinetic and potential energy operator). $\psi(u,t)$ denotes the (complex-valued) quantum wave function at position $u$ at time $t$, and the square modulus of the wave function $|\psi(u,t)|^2$ indicates the probability density of a particle. With the quantum system defined by $\psi(u,t)$, a quantum state $|\psi(t)\rangle$ refers to the superposition of all the states, i.e., $|\psi(t)\rangle = \int \psi(u,t)|u\rangle du$, where $|u\rangle$ is the position state (a basis vector for representing the quantum state). One can recover the components of the quantum state by the inner product between quantum state vector $|\psi(t)\rangle$ and position vector $|u\rangle$, i.e., $\psi(u,t) = \langle u|\psi(t)\rangle$. In a simple quantum system where the Hamiltonian $\mathcal{H}$ is time-independent, the solution to the Schrödinger's equation can be written in terms of quantum state as $|\psi(t)\rangle = e^{-i\mathcal{H}t}|\psi(0)\rangle$.

On a graph with $n$ nodes, $|\psi(t)\rangle \in \mathbb{C}^n$ can be interpreted as the state vector of particles across all nodes, and the position state refers to the node set. In a system without a potential and thus the potential energy term is zero, the Hamiltonian reduces to the negative Laplacian, and the Schrödinger's equation reduces to $i\frac{\partial|\psi(t)\rangle}{\partial t} = -\mathbf{L}|\psi(t)\rangle$. The eigenvectors of $\mathbf{L}$, denoted as $\{|\phi_i\rangle\}_{i=1}^n$ provides a complete orthogonal basis, which can be treated as the position basis so that one can express the solution as $|\psi(t)\rangle = \sum_{k=1}^n c_k e^{i\lambda_k t}|\phi_k\rangle$, where $\lambda_k$ is the $k$-th eigenvalue and $c_k = \langle\psi(0)|\phi_k\rangle$. Instead of working with the solution which involves the complex unit, Markovich (2023) define a kernel that models the average overlap between any two nodes $i, j$ as the inner product between $|\psi(t)\rangle_i$ and $|\psi(t)\rangle_j$, which is $\sum_{k,l=1}^n c_k^* c_l \langle\phi_k\rangle_i^* |\phi_l\rangle_j$. To engineer observation operators as a spectral filter, the paper further leverages a Gaussian filter $\mathcal{P} = \sum_{k=1}^n e^{-(\lambda_k - \mu)^2/2\sigma^2}$, which leads to the proposed quantum diffusion kernel (QDC) $\mathbf{Q} \in \mathbb{R}^{n \times n}$, where

$$\text{QDC}: \quad \mathbf{Q}_{i,j} = \sum_{k=1}^n e^{-(\lambda_k - \mu)^2/2\sigma^2}|\phi_k\rangle_i^* |\phi_k\rangle_j$$

The kernel matrix $\mathbf{Q}$ is interpreted as the transition matrix between the nodes and hence can be supplemented for any message-passing-based graph neural networks. Hence the resulting quantum diffusion corresponds to the anisotropic diffusion where the diffusivity is given by the quantum diffusion kernel. The kernel matrix can be further sparsified either using a threshold or KNN. The kernel allows non-local message passing due to the quantum interference across all the position states (i.e., $\mathbf{Q}_{i,j}$ is computed with all the states). Further a multi-scale variant of quantum diffusion is proposed that combines the propagation from quantum diffusion and standard graph diffusion.

**Time derivative diffusion.** Another work (Behmanesh et al., 2023) introduces a non-local message passing scheme by combining local message passing with a learnable-timestep heat kernel. Recall the heat diffusion follows $\frac{\partial \mathbf{X}}{\partial t} = -\mathbf{L}\mathbf{X}$, where its solution is given by the heat kernel as $\mathbf{X}(t) = \exp(-t\mathbf{L})\mathbf{X}(0)$. One can generalize the heat kernel to capture the transition between any two states in time, i.e., $\mathbf{X}(t) = \exp(-(t-s)\mathbf{L})\mathbf{X}(s)$. Instead of setting a pre-defined $t, s$, the paper parameterizes the heat kernel as $\exp(-t_\theta\mathbf{L})$ where $t_\theta$ is the learnable timestep in order to adapt the diffusion range to different types of a dataset. Further, to simultaneously account for local message passing, the paper combines the adjacency matrix $\mathbf{A}$ with learned heat kernel, which leads to the proposed dynamics as $\mathbf{X}(t) = \mathbf{A}\exp(-t_\theta\mathbf{L})\mathbf{X}(s)$. This corresponds to the following continuous dynamics (up to some normalizing constants):

$$\text{TIDE}: \quad \frac{\partial \mathbf{X}}{\partial t} = \text{div}(\mathbf{A}\exp(-t_\theta\mathbf{L}) \cdot \nabla\mathbf{X}).$$

It is clear that when $t_\theta = 0$, the model reduces to the classic GCN. The learnability of $t_\theta$ ensures the model is flexible to capture both local and multi-hop communication.

**Hypo-elliptic diffusion.** In (Toth et al., 2022), non-local and higher-order information is captured via the so-called *hypo-elliptic diffusion*, which is based on a tensor-valued Laplacian that aggregates the entire

trajectories of random walks on graphs. The sequential nature of the path can be characterized with the free (associative) algebra, which lifts the sequence injectively to a vector space with non-commutative multiplication (i.e., an algebra). More formally, an algebra $H$ over $\mathbb{R}^c$ can be realized as a sequence of tensors with increasing order, i.e., $H := \{\mathbf{v} = (\mathbf{v}^0, \mathbf{v}^1, \mathbf{v}^2, ...): \mathbf{v}^m \in (\mathbb{R}^c)^{\otimes m}\}$, where $(\mathbb{R}^c)^{\otimes m}$ denotes the space of $m$-order tensors. For example, $(\mathbb{R}^c)^{\otimes 0} \equiv \mathbb{R}, (\mathbb{R}^c)^{\otimes 1} \equiv \mathbb{R}^c$. The scalar multiplication and vector addition of $H$ is defined according to $\lambda\mathbf{v} := (\lambda\mathbf{v}^m)_{m\geq 0}$ for $\lambda \in \mathbb{R}$ and $\mathbf{v} + \mathbf{w} := (\mathbf{v}^m + \mathbf{w}^m)_{m\geq 0}$. The algebra multiplication of $H$ is given by $\mathbf{v} \cdot \mathbf{w} := \left(\sum_{k=0}^m \mathbf{v}^k \otimes \mathbf{w}^{m-k}\right)_{m\geq 0}$. $H$ can be further made into a Hilbert space with the chosen inner product $\langle\mathbf{v}, \mathbf{w}\rangle := \sum_{m\geq 0}\langle\mathbf{v}^m, \mathbf{w}^m\rangle^m$ where $\langle\cdot, \cdot\rangle^m$ denotes the classic inner product on the tensor space $(\mathbb{R}^c)^{\otimes m}$.

With the properly defined algebra, one can lift a sequence to such space, thus summarizing the full details of its trajectory. Specifically, denote the space of sequences in $\mathbb{R}^c$ as $\text{Seq}(\mathbb{R}^c) := \bigcup_{k=0}^{\infty}(\mathbb{R}^c)^{k+1}$, where $(\mathbb{R}^c)^{k+1}$ denotes the $k+1$-product space of $\mathbb{R}^c$. A sequence, denoted as $\mathbf{x} = (\mathbf{x}^0, \mathbf{x}^1, ..., \mathbf{x}^k) \in (\mathbb{R}^c)^{k+1}$ is an element of the sequence space $\text{Seq}(\mathbb{R}^c)$. Let $\varphi : \mathbb{R}^c \to H$ be an algebra lifting, which allows to define a sequence feature map $\tilde{\varphi}(\mathbf{x}) = \varphi(\mathbf{x}^0) \cdot \varphi(\mathbf{x}^1 - \mathbf{x}^0) \cdots \varphi(\mathbf{x}^k - \mathbf{x}^{k-1}) \in H$. One example of such injective map is the tensor exponential given by $\varphi(\mathbf{u}) = \exp_{\otimes}(\mathbf{u}) := \left(\frac{\mathbf{u}^{\otimes m}}{m!}\right)_{m\geq 0}$ for $\mathbf{u} \in \mathbb{R}^c$, where $x^{\otimes m} := \underbrace{x \otimes x \cdots \otimes x}_{m}$. Such a feature map is able to summarize the entire sequence path up to step $k$.

On a graph, let $\mathbf{x}_i^k \in \mathbb{R}^c$ denote the signals at node $i \in \mathcal{V}$ at diffusion step $k \geq 0$. Instead of focusing on the signals at a particular timestep $k$, the paper leverages the sequence map to capture the entire past trajectory of the diffusion process through $\tilde{\varphi}(\mathbf{x}_i) \in H$ where $\mathbf{x}_i := (\mathbf{x}_i^0, \mathbf{x}_i^1, ..., \mathbf{x}_i^k)$. The corresponding diffusion process requires a tensor adjacency matrix, $\widetilde{\mathbf{A}} \in H^{n \times n}$ with the entries $\widetilde{\mathbf{A}}_{i,j} = \varphi(\mathbf{x}_j^0 - \mathbf{x}_i^0) \in H$ if $(i, j) \in \mathcal{E}$ and 0 otherwise. The associated Laplacian $\widetilde{\mathbf{L}}$ can be defined accordingly. For example, the random walk Laplacian has entries $\widetilde{\mathbf{L}}_{i,i} = 1$ and $\widetilde{\mathbf{L}}_{i,j} = -\frac{1}{\deg_i}\varphi(\mathbf{x}_j^0 - \mathbf{x}_i^0)$ if $(i, j) \in \mathcal{E}$ and 0 otherwise. The classic graph heat diffusion is generalized to hypo-elliptic graph diffusion as

$$\text{G2TN}: \quad \frac{\partial\tilde{\varphi}(\mathbf{X})}{\partial t} = -\widetilde{\mathbf{L}}\tilde{\varphi}(\mathbf{X}),$$

where we let $\tilde{\varphi}(\mathbf{X}) := [\tilde{\varphi}(\mathbf{x}_1), ..., \tilde{\varphi}(\mathbf{x}_n)] \in H^n$. The multiplication of $\widetilde{\mathbf{L}}$ and $\tilde{\varphi}(\mathbf{X})$ is defined over the space of algebra $H$ (similarly as how classic matrix multiplication works). In the case of random walk Laplacian, the paper verifies that the solution of the hypo-elliptic diffusion summarizes the entire random walk histories of each node, in contrast to the snapshot state at each timestep given by the classic diffusion equation. Notice here instead of working with node signals $\mathbf{x}$ directly, the diffusion concerns all the node signals along the trajectory, i.e., $\varphi(\mathbf{x})$. Thus $\tilde{\varphi}(\mathbf{x})$ is more expressive compared to $\mathbf{x}$. The paper further adapts the attention mechanism to define a weighted hypo-elliptic adjacency as $\widetilde{\mathbf{A}}_{i,j} = a(\mathbf{x}_i^0, \mathbf{x}_j^0)\varphi(\mathbf{x}_j^0 - \mathbf{x}_i^0)$, which correspondingly defines an anisotropic hypo-elliptic diffusion.

### 3.4 Diffusion with external forces

Most of the aforementioned processes are only controlled by a single mechanism, either diffusion or oscillation. This section discusses systems that impart external forces to the diffusion dynamics, such as *convection*, *advection*, and *reaction*. In particular, convection and advection are widely known in physical sciences that describe how the mass transports as a result of the movement of the underlying fluid. Such a process is characterized by $\frac{\partial x}{\partial t} = -\text{div}(\mathbf{v}x)$ where $\mathbf{v}$ represents the velocity field of the fluid motion. Reaction process is more general and often found in chemistry where chemical substance interacts with each other and leads to a change of mass and substance, i.e., $\frac{\partial x}{\partial t} = r(x)$, for some reaction function $r(\cdot)$. Other mechanisms such as gating, reverse diffusion and anti-symmetry have also been exploited in literature to modulate and control the diffusion dynamics.

**Convection-diffusion.** Convection-diffusion dynamics combines the convection with diffusion process in which mass not only transports but also disperse in space. Zhao et al. (2023b) generalize the convection-diffusion equation (CDE) to graphs as

$$\text{CDE}: \quad \frac{\partial\mathbf{X}}{\partial t} = \text{div}(\mathbf{A}(\mathbf{X}) \cdot \nabla\mathbf{X}) + \text{div}(\mathbf{V} \circ \mathbf{X})$$

where $\mathbf{V}$ denotes a velocity field and $(\mathbf{V} \circ \mathbf{X})_{i,j} := \mathbf{V}_{i,j} \odot \mathbf{x}_j$ with $\odot$ representing the elementwise product. In particular, Zhao et al. (2023b) define $\mathbf{V}_{i,j} = \sigma(\mathbf{W}(\mathbf{x}_j - \mathbf{x}_i)) \in \mathbb{R}^c$ for some nonlinear activation $\sigma(\cdot)$ and learnable matrix $\mathbf{W}$. Such a choice is motivated from the heterophilic graphs where neighbouring nodes exhibit diverse features. Hence $\mathbf{V}_{i,j}$ captures the dissimilarity between the nodes, which ultimately guides the diffusion process. Accordingly, $\mathrm{div}(\mathbf{V} \circ \mathbf{X})_i = \sum_{j:(i,j)\in\mathcal{E}} \mathbf{V}_{i,j} \odot \mathbf{x}_j$ measures the flow of density at node $i$ in the direction of $\mathbf{V}_{i,j}$. The nonlinear parameterization of the gradient further enhances the dynamics to adapt to different graphs with varying homophily levels.

**Reaction-diffusion.** A more general reaction-diffusion process is considered in (Choi et al., 2023):

$$\text{GREAD}: \quad \frac{\partial \mathbf{X}}{\partial t} = \mathrm{div}(\mathbf{A}(\mathbf{X}) \cdot \nabla \mathbf{X}) + R(\mathbf{X}),$$

where $R(\mathbf{X})$ is the reaction term, and proper choice of $R(\cdot)$ recovers many existing works as special cases. For example, the Fisher reaction (Fisher, 1937) is given by $R(\mathbf{X}) = \kappa \mathbf{X} \odot (\mathbf{1} - \mathbf{X})$, which can be used to model the spread of biological populations where $\kappa$ represents the intrinsic growth rate. Other reaction processes include Allen-Cahn (Allen & Cahn, 1979) $R(\mathbf{X}) = \mathbf{X} \odot (1 - \mathbf{X} \odot \mathbf{X})$ and Zeldovich (Gilding & Kersner, 2004) $R(\mathbf{X}) = \mathbf{X} \odot (\mathbf{X} - \mathbf{X} \odot \mathbf{X})$. Apart from the physics informed choices of reaction term, Choi et al. (2023) also consider $R(\mathbf{X}) = \mathbf{X}^0$, which follows GRAND++, CGNN to incorporate a source term, and also proposes several high-pass reaction term based on graph structure, aiming to induce a sharpening effect. For example, the blurring-sharpening reaction is defined as $R(\mathbf{X}) = (\mathbf{A}(\mathbf{X}) - \mathbf{A}(\mathbf{X})^2)\mathbf{X}$, which corresponds to performing a low-pass filter $\mathbf{A}(\mathbf{X}) - \mathbf{I}$ followed by a high-pass filter $\mathbf{I} - \mathbf{A}(\mathbf{X})$. The paper also considers two learnable coefficients $\alpha, \beta$ to control the emphasis on diffusion and reaction terms respectively, i.e., $\alpha \, \mathrm{div}(\mathbf{A}(\mathbf{X}) \cdot \nabla \mathbf{X}) + \beta R(\mathbf{X})$.

A closely related work (Wang et al., 2023) adopts the Allen-Cahn reaction term for the reaction-diffusion process. Further, the paper allows negative diffusion coefficients, which is able to induce a repulsive force between the nodes:

$$\text{ACMP}: \quad \frac{\partial \mathbf{X}}{\partial t} = \mathrm{div}((\mathbf{A}(\mathbf{X}) - \mathbf{B}) \cdot \nabla \mathbf{X}) + \mathbf{X} \odot (\mathbf{1} - \mathbf{X} \odot \mathbf{X})$$

where $\mathbf{B} > 0$ is a bias term controlling the strength and direction of message passing. This allows $\mathbf{A}(\mathbf{X}) - \mathbf{B}$ to model the interactive forces and can become negative. Further, the first term of ACMP corresponds to the gradient flow of a pseudo Dirichlet energy given by $\sum_{(i,j)\in\mathcal{E}}(\mathbf{A}_{i,j} - \mathbf{B}_{i,j})\|\mathbf{x}_i - \mathbf{x}_j\|^2$ where we ignore the dependence of $\mathbf{A}$ on $\mathbf{X}$ for the time being. This suggests, when $\mathbf{A}_{i,j} - \mathbf{B}_{i,j} > 0$, node $i$ is attracted by node $j$ by minimizing the difference between $\mathbf{x}_i$ and $\mathbf{x}_j$ and when $\mathbf{A}_{i,j} - \mathbf{B}_{i,j} < 0$, node $i$ is repelled by node $j$. It is noticed that the presence of negative weights can cause the energy to be unbounded and thus the dynamics may not be convergent. To resolve the issue, the paper considers an external potential, namely the double-well potential $(\delta/4)\sum_{i\in\mathcal{V}}(1 - \|\mathbf{x}_i\|^2)^2$. ACMP is indeed derived as the gradient flow of the pseudo Dirichlet energy combined with the double-well potential. Theoretically, the Dirichlet energy $\mathcal{E}_{\mathrm{dir}}$ of ACMP evolution is upper bounded due to the Allen-Cahn reaction term as well as lower bounded due to the repulsive forces. Hence, the system remains stable while avoiding oversmoothing. For practical implementation, ACMP sets $\mathbf{B}_{i,j} = \beta > 0$, a tunable hyperparameter for simplicity of optimization. Two channel-wise coefficient vectors are added to balance the diffusion and reaction similarly in (Choi et al., 2023). Unlike in ACMP (Wang et al., 2023) where the repulsive components $\mathbf{B}_{i,j}$ are treated as hyperparameters, a recent work (Shi et al., 2024) proposes a more principled strategy for incorporating negative weights through augmented nodes with label information.

In addition, the idea of incorporating repulsive force in the message passing has also been explored in (Lv et al., 2023). The work proposes to view the message passing mechanism on graphs in the framework of opinion dynamics. The work explores the notion of *bounded confidence* from the Hegselmann-Krause (HK) model (Hegselmann & Krause, 2002) where only similar opinions (up to some cut-off threshold) are exchanged. This motivates the following dynamics on graphs that separates messages according to the similarity level of graph signals:

$$\text{ODNet}: \quad \frac{\partial \mathbf{X}}{\partial t} = \mathrm{div}_{\mathcal{V}\times\mathcal{V}}(\Phi(\mathbf{A}(\mathbf{X})) \cdot \nabla \mathbf{X}) + R(\mathbf{X}),$$

where $\text{div}_{\mathcal{V} \times \mathcal{V}}$ defines message aggregation over the complete graph. $\Phi(\mathbf{A}(\mathbf{X}))$ is an elementwise scalar-valued function (called influence function) on diffusivity and is required to be non-decreasing. In (Lv et al., 2023), $\Phi(\cdot)$ is chosen to be piecewise linear as follows.

$$\Phi(s) = \begin{cases} \mu s, & \text{if } s > \epsilon_2, \\ s, & \text{if } \epsilon_1 \leq s \leq \epsilon_2, \\ \nu(1-s), & \text{otherwise} \end{cases}$$

where $\mu > 0$ and $\nu \leq 0$ are hyperparameters. In addition, $\epsilon_1, \epsilon_2$ defines the influence regions (which resembles bounded confidence in HK model). Because $\nu$ can be negative, it is able to induce repulsive forces by separating the node representations. Empirically, the work chooses $\nu = 0$ for homophilic graphs and $\nu < 0$ for heterophilic graphs. It is noticed that when $\nu < 0$, the message can propagate even for unconnected nodes in the case of $\nu < 0$.

**Advection-diffusion-reaction.** ADR-GNN (Eliasof et al., 2023) further adds an explicit advection term on top of the reaction-diffusion process considered in (Choi et al., 2023).

$$\text{ADR-GNN}: \quad \frac{\partial \mathbf{X}}{\partial t} = \text{div}(\mathbf{A}(\mathbf{X}) \cdot \nabla \mathbf{X}) + \text{div}(\mathbf{V} \circ \mathbf{X}) + R(\mathbf{X}).$$

The work considers homogeneous diffusion with channel scaling, i.e., $\text{div}(\mathbf{A}(\mathbf{X}) \cdot \nabla \mathbf{X}) = -\mathbf{LX}\text{diag}(\boldsymbol{\theta})$, where $\boldsymbol{\theta} \in \mathbb{R}^c$ is the channel-wise scaling factor. In contrast to CDE (Zhao et al., 2023b), the advection term concerns two directional velocity fields $\mathbf{V}_{i,j}, \mathbf{V}_{j,i} \in \mathbb{R}^c$, which measures both in-flow and out-flow of density, with $(\mathbf{V} \circ \mathbf{X})_{i,j} := \mathbf{V}_{j,i} \odot \mathbf{x}_j - \mathbf{V}_{i,j} \odot \mathbf{x}_i$. By further ensuring channel-wise row stochasticity of both $\mathbf{V}_{i,j}$ and $\mathbf{V}_{j,i}$, $\mathbf{V}_{i,j} \odot \mathbf{x}_i$ is interpreted as the mass of node $j$ to be transported to node $i$. Thus the advection term, given by $\text{div}(\mathbf{V} \circ \mathbf{X})_i = \sum_{j:(i,j)\in\mathcal{E}} \mathbf{V}_{j,i} \odot \mathbf{x}_j - \mathbf{x}_i$, quantifies the net flow of density at node $i$. The reaction term $R(\mathbf{X})$ is parameterized by additive and multiplicative MLPs with a source term, $R(\mathbf{X}) = \sigma(\mathbf{XW}_1 + \tanh(\mathbf{XW}_2) \odot \mathbf{X} + \mathbf{X}^0 \mathbf{W}_3)$. Different to previous works, the paper considers operator splitting for discretizing the continuous dynamics, i.e., by separating the propagation of the three processes. Such a scheme allows separate treatment and analysis of each process. Particularly, Eliasof et al. (2023) demonstrate the mass preserving property and stability of the advection operator. Empirically, ADR-GNN has shown promising performance for modelling not only the static graphs but also spatial temporal graphs where advection-diffusion-reaction process has been successful (Fiedler & Scheel, 2003).

**Gating.** It has been shown that controlling the speed of diffusion through convection/advection term is able to counteract the smoothing process. Rusch et al. (2023b) adopt a different strategy by explicitly modelling a gating function on the diffusion:

$$\text{G}^2: \quad \frac{\partial \mathbf{X}}{\partial t} = \mathbf{T}(\mathbf{X}) \odot \text{div}(\mathbf{A}(\mathbf{X}) \cdot \nabla \mathbf{X})$$

where $\mathbf{T}(\mathbf{X}) \in [0,1]^{n \times c}$ collects the rate of speed for each node and across each channel. Specifically, the rates depend on the graph gradient as $T(\mathbf{X})_{i,k} = \tanh(\sum_{j\in\mathcal{N}_i} |\hat{\mathbf{X}}_{j,k} - \hat{\mathbf{X}}_{i,k}|^p), p > 0$ where $\hat{\mathbf{x}}_i = \sum_{j\in\mathcal{N}_i} \widehat{\mathbf{A}}(\mathbf{X})_{i,j}\mathbf{x}_j$ and $\widehat{\mathbf{A}}(\mathbf{X})$ is another message aggregation. Conceptually, the gating rates $T(\mathbf{X})_{i,:}$ for node $i$ depend on the channel-wise graph gradients to all its neighbours. The use of $\tanh(\cdot)$ ensures when $\sum_{j\in\mathcal{N}_i} |\hat{\mathbf{X}}_{j,k} - \hat{\mathbf{X}}_{i,k}|^p \to 0$, the rate $T(\mathbf{X})_{i,k}$ vanishes at a faster rate. This correspondingly shuts down the update for node $i$ and thus avoid oversmoothing. The paper also considers more general choices of coupling functions in place of $\text{div}(\mathbf{A}(\mathbf{X}) \cdot \nabla \mathbf{X})$ where nonlinearity is added.

The idea of gating has been similarly explored in DeepGRAND (Nguyen et al., 2023), which utilizes a channel-wise scaling factor $\langle \mathbf{X} \rangle^p \in \mathbb{R}^{n \times d}$ in place of $T(\mathbf{X})$ where $\langle \mathbf{X} \rangle^p_{:,k} = \|\mathbf{X}_{:,k}\|^p \mathbf{1}_n$. The dynamics also incorporates a perturbation to the diffusivity as $\mathbf{A}(\mathbf{X}) - (1 + \epsilon)\mathbf{I}$. The scaling factor and perturbation help regulate the convergence of node features so that the node features neither explodes nor converges too fast to the steady state.

**Reverse diffusion.** Similar to the idea in (Choi et al., 2023) that simultaneously accounts for low-pass and high-pass filters, Shao et al. (2023) introduce a reverse diffusion process based on the heat kernel. When coupled with heat diffusion, it leads to a process that accommodates both smoothing and sharpening effects:

$$\text{MHKG}: \quad \frac{\partial \mathbf{X}}{\partial t} = \big(\text{diag}(\boldsymbol{\theta}_1)\exp(f(\widehat{\mathbf{L}})) + \text{diag}(\boldsymbol{\theta}_2)\exp(g(\widehat{\mathbf{L}})) - \mathbf{I}\big)\mathbf{X},$$

where $\boldsymbol{\theta}_1, \boldsymbol{\theta}_2$ are learnable filters and $f, g$ are scalar-valued functions defined over the eigenvalues of $\widehat{\mathbf{L}}$, e.g., $f(\widehat{\mathbf{L}}) := \mathbf{U}f(\boldsymbol{\Lambda})\mathbf{U}^\top$ where $f(\boldsymbol{\Lambda}) = \text{diag}([f(\lambda_i)]_{i=1}^n)$, with $\boldsymbol{\Lambda}$ being the diagonal matrix collecting the eigenvalues and $\mathbf{U}$ collecting the eigenvectors of the normalized Laplacian $\widehat{\mathbf{L}}$. In particular, $f, g$ are assumed to be opposite in terms of monotonicity. In the simplest case, suppose $f(\widehat{\mathbf{L}}) = -\widehat{\mathbf{L}}$ and $g(\widehat{\mathbf{L}}) = \widehat{\mathbf{L}}$, then the two terms in MHKG correspond to the heat kernel and its reverse. The filtering coefficients $\boldsymbol{\theta}_1, \boldsymbol{\theta}_2$ controls the relative dominance of the two terms, where the former smooths while the latter sharpens the signals.

**Anti-symmetry.** In (Gravina et al., 2023), a stable and non-dissipative system is proposed by imposing the additional anti-symmetric structure for channel mixing matrix as

$$\text{A-DGN}: \quad \frac{\partial \mathbf{X}}{\partial t} = \mathbf{X}(\mathbf{W} - \mathbf{W}^\top) + F_{\mathcal{G}}(\mathbf{X}),$$

where we omit the nonlinearity and a bias term to show only the driving factors. Here, $F_{\mathcal{G}}(\mathbf{X})$ is a coupling function similar as in (Rusch et al., 2022), such as a simple homogeneous message passing $F_{\mathcal{G}}(\mathbf{X})_i = \sum_{j \in \mathcal{N}_i} \mathbf{V}\mathbf{x}_j$ for some weight matrix $\mathbf{V}$ or the one with attention mechanism (Veličković et al., 2018). The incorporation of anti-symmetry constraint for the channel mixing renders the system to be stable and non-dissipative, both due to the fact that the Jacobian of dynamics has pure imaginary eigenvalues, i.e., all real parts of the eigenvalues are zero. This suggests the solutions to the system remain bounded under perturbation of initial conditions, which concludes the stability of the evolution. In addition, the zero real part of the eigenvalues suggests the sensitivity of the node signals to its initial values, i.e., the magnitude of $\frac{\partial \mathbf{x}_i(t)}{\partial \mathbf{x}_i(0)}, \forall i, t$ stays constant throughout the dynamics. This result infers that oversmoothing in the limit does not occur as the final state still depends on the initial conditions as $\lim_{t\to\infty} \frac{\partial \mathbf{x}_i(t)}{\partial \mathbf{x}_i(0)} \neq 0$. Further, this also suggests the magnitude of $\frac{\partial L_{\text{loss}}}{\partial \mathbf{x}_i(0)}$ remains unchanged over time and hence gradient vanishing or explosion is avoided during backpropagation. This allows the dynamics to be propagating to the limit and capture long-range interactions without facing the issue of oversmoothing or training instability.

### 3.5 Geometry-underpinned dynamics

Previous sections have viewed graphs from its trivial topology, and standard dynamics on graphs amounts to propagating information from node to edge space and back, only utilizing the connectivity between nodes. In fact, graphs can often be viewed as discrete approximations of more general topological spaces such as Riemannian manifolds, which possess complex continuous geometries. In this section, we show how dynamics on graphs can be underpinned with additional geometric structure, such as sheaves and stalks in (Bodnar et al., 2022), and cliques and cochains in (Gruber et al., 2023).

**Sheaf diffusion.** Hansen & Gebhart (2020); Barbero et al. (2022a) and Bodnar et al. (2022) leverage *cellular sheaf theory* to endow a geometric structure for graphs. Specifically, each node $i \in \mathcal{V}$ and edge $e_{ij} = \{i, j\} \in \mathcal{E}$ (undirected) is equipped with a vector space structure $\mathcal{F}(i), \mathcal{F}(e_{ij})$, with a linear map $\mathcal{F}_{i \vartriangleleft e_{ij}} : \mathcal{F}(i) \to \mathcal{F}(e_{ij})$ (called restriction map) that connects the node to edge spaces. Its adjoint operator $\mathcal{F}_{i \vartriangleleft e_{ij}}^\top : \mathcal{F}(e_{ij}) \to \mathcal{F}(i)$ does the reverse. The direct sum of all vector spaces of nodes is called the space of 0-cochains, denoted by $C^0(\mathcal{G}; \mathcal{F}) := \bigoplus_{i \in \mathcal{V}} \mathcal{F}(i)$. Suppose, without loss of generality, that all vector spaces $\mathcal{F}(i), \mathcal{F}(e_{ij})$ are $d$-dimensional, we can represent $\mathcal{F}_{i \vartriangleleft e_{ij}}$ as a $d \times d$ matrix. Further, suppose $\mathbf{x} \in C^0(\mathcal{G}; \mathcal{F})$, then each $\mathbf{x}_i \in \mathbb{R}^d$ and $\mathbf{x} \in \mathbb{R}^{nd}$ by stacking the feature vectors across all nodes.

Under the construction of vector spaces on nodes and edges, the concepts of graph gradient and divergence require to utilize the restriction maps because the vector spaces are not directly comparable. That is, the graph gradient (also known as the co-boundary map) is defined as $(\nabla_{\mathcal{F}}\mathbf{x})_{i,j} := \mathcal{F}_{j \vartriangleleft e_{ij}}\mathbf{x}_j - \mathcal{F}_{i \vartriangleleft e_{ij}}\mathbf{x}_i$, where

the restriction maps transport the signals to a common disclosure space. The graph divergence is thus similarly defined as $(\text{div}_{\mathcal{F}}\mathbf{G})_i = \sum_{j \in \mathcal{N}_i} \boldsymbol{\mathcal{F}}_{i \triangleleft e_{ij}}^{\top} \mathbf{G}_{e_{ij}}$, for $\mathbf{G}_{e_{ij}} \in \mathcal{F}(e_{ij})$. This leads to the definition of sheaf Laplacian as $\mathbf{L}_{\mathcal{F}}(\mathbf{x})_i = \text{div}_{\mathcal{F}}(\nabla_{\mathcal{F}}\mathbf{x})_i = \sum_{j \in \mathcal{N}_i} \boldsymbol{\mathcal{F}}_{i \triangleleft e_{ij}}^{\top}(\boldsymbol{\mathcal{F}}_{i \triangleleft e_{ij}}\mathbf{x}_i - \boldsymbol{\mathcal{F}}_{j \triangleleft e_{ij}}\mathbf{x}_j)$. The sheaf Laplacian is an $nd \times nd$ block positive semi-definite matrix, with the diagonal blocks $(\mathbf{L}_{\mathcal{F}})_{i,i} = \sum_{j \in \mathcal{N}_i} \boldsymbol{\mathcal{F}}_{i \triangleleft e_{ij}}^{\top}\boldsymbol{\mathcal{F}}_{i \triangleleft e_{ij}}$ and off-diagonal blocks $(\mathbf{L}_{\mathcal{F}})_{i,j} = -\boldsymbol{\mathcal{F}}_{i \triangleleft e_{ij}}^{\top}\boldsymbol{\mathcal{F}}_{j \triangleleft e_{ij}}$. A symmetrically normalized sheaf Laplacian can be similarly computed by $\mathbf{D}_{\mathcal{F}}^{-1/2}\mathbf{L}_{\mathcal{F}}\mathbf{D}_{\mathcal{F}}^{-1/2}$ with $\mathbf{D}_{\mathcal{F}}$ is the block diagonal of $\mathbf{L}_{\mathcal{F}}$. The corresponding sheaf diffusion process is

$$\frac{\partial \mathbf{X}}{\partial t} = \text{div}_{\mathcal{F}}(\nabla_{\mathcal{F}}\mathbf{X}) = -\mathbf{L}_{\mathcal{F}}\mathbf{X}$$

where here $\mathbf{X} \in \mathbb{R}^{(nd) \times c}$, where $c$ is the feature channels and $\text{div}_{\mathcal{F}}, \nabla_{\mathcal{F}}, \mathbf{L}_{\mathcal{F}}$ are applied channel-wise. The Sheaf diffusion turns out to be the gradient flow of the sheaf Dirichlet energy $\text{tr}(\mathbf{X}^{\top}\mathbf{L}_{\mathcal{F}}\mathbf{X})$, which measures the smoothness of signals in the disclosure space. For practical settings where sheaf structure is unavailable, one can construct such a feature through input embedding. It is worth highlighting that, when $d = 1$, the sheaf Laplacian reduces to the classic graph Laplacian and the sheaf diffusion becomes the heat diffusion. In (Bodnar et al., 2022), a variety of restriction maps are constructed, which leads to dynamics flexible enough to handle different types of graphs and avoid oversmoothing. The paper also develops a general framework for learning the sheaf Laplacian from the features and include channel mixing and nonlinearity to increase the expressive power:

$$\text{NSD}: \quad \frac{\partial \mathbf{X}}{\partial t} = -\sigma(\mathbf{L}_{\mathcal{F}(\mathbf{X})}(\mathbf{I} \otimes \mathbf{W}_1)\mathbf{X}\mathbf{W}_2)$$

where $\mathbf{W}_1 \in \mathbb{R}^{d \times d}$ transforms the feature vectors and $\mathbf{W}_2 \in \mathbb{R}^{c \times c'}$ mixes the channels and $\otimes$ denotes the Kronecker product. $\mathbf{L}_{\mathcal{F}(\mathbf{X})}$ is parameterized via a matrix-valued function on the current feature values.

In (Bodnar et al., 2022), the sheaf Laplacian is parameterized by $\boldsymbol{\mathcal{F}}_{i \triangleleft e_{ij}} = \sigma(\mathbf{V}[\mathbf{x}_i \| \mathbf{x}_j])$ where $\cdot \| \cdot$ denotes concatenation. A follow-up work (Barbero et al., 2022b) devises a sheaf attention mechanism to further enhance the diffusion process. Let $\mathbf{A}(\mathbf{X}) \in \mathbb{R}^{n \times n}$ be a matrix of learnable attention coefficients (the same as in GAT (Veličković et al., 2018)), and let $\widehat{\mathbf{A}}(\mathbf{X}) \coloneqq \mathbf{A}(\mathbf{X}) \otimes \mathbf{1}_{d \times d}$ that assigns uniform attention coefficients for each feature dimension within a vector space. The attentive sheaf diffusion (SheafAN) is introduced as $\frac{\partial \mathbf{X}}{\partial t} = (\widehat{\mathbf{A}}(\mathbf{X}) \odot \mathbf{A}_{\mathcal{F}(\mathbf{X})} - \mathbf{I})\mathbf{X}$, where $(\mathbf{A}_{\mathcal{F}(\mathbf{X})})_{i,j} = \boldsymbol{\mathcal{F}}_{i \triangleleft e_{ij}}^{\top}\boldsymbol{\mathcal{F}}_{j \triangleleft e_{ij}}$ is the sheaf adjacency matrix with self-loop, i.e., $e_{ii} \in \mathcal{E}$. A second-order sheaf PDE (NSP) is proposed in (Suk et al., 2022) using the wave equation as $\frac{\partial^2 \mathbf{X}}{\partial t^2} = \text{div}_{\mathcal{F}}(\nabla_{\mathcal{F}}\mathbf{X})$.

**Bracket dynamics.** Gruber et al. (2023) propose to use *geometric brackets* that implicitly parameterize dynamics on graphs that satisfy certain properties while equipping graphs with higher-order structures. The formulation requires concepts from structure-preserving bracket-based dynamics and exterior calculus. In general, for a state variable $x$, its dynamics can be given by some combination of reversible and irreversible brackets. The *reversible* bracket (also known as Poisson bracket) is denoted by $\{A, E\} \coloneqq \langle \frac{\partial A}{\partial x}, \tilde{L}\frac{\partial E}{\partial x} \rangle$ for some skew-symmetric operator $\tilde{L}$ * and some inner product $\langle \cdot, \cdot \rangle$. The reversibility is a result of energy conservation. The *irreversible* bracket is defined by $[A, E] \coloneqq \langle \frac{\partial A}{\partial x}, M\frac{\partial E}{\partial x} \rangle$ for some (either positive or negative) semi-definite operator $M$. The irreversibility describes the loss of energy from the system due to friction or dissipation. The double bracket $\{\{A, E\}\} \coloneqq \langle \frac{\partial A}{\partial x}, \tilde{L}^2\frac{\partial E}{\partial x} \rangle$ is an irreversible bracket.

For simplicity, the paper considers $A = x$ and one can simplify the brackets as $[x, E] = M\frac{\partial E}{\partial x}$ and $\{x, E\} = \tilde{L}\frac{\partial E}{\partial x}$. The paper considers four different types of dynamics leveraging both reversible and irreversible brackets:

$$
\begin{aligned}
\text{Hamiltonian}: &\quad \frac{\partial x}{\partial t} = \{x, E\}; \\
\text{Gradient}: &\quad \frac{\partial x}{\partial t} = -[x, E]; \\
\text{Double bracket}: &\quad \frac{\partial x}{\partial t} = \{x, E\} + \{\{x, E\}\}; \\
\text{Metriplectic}: &\quad \frac{\partial x}{\partial t} = \{x, E\} + [x, S],
\end{aligned}
\tag{3}
$$

where $E(x)$ is referred to as the energy of the state and $S(x)$ is the entropy. The dynamics of each process captures fundamentally different systems and has natural substance in physics. The Hamiltonian dynamics

---

*Here, in order not to be confused with the Laplacian $L$ used in previous discussions, we use $\tilde{L}$.

leads to a complete, isolated system in the sense that no energy is lost to the external environment. In contrast, both the gradient and double bracket dynamics are incomplete where the energy is lost through the process. Metriplectic dynamics is complete by further requiring the degeneracy conditions $\tilde{L}\frac{\partial S}{\partial x} = M\frac{\partial E}{\partial x} = 0$. These conditions ensure the conservation of energy, i.e., $\frac{\partial E}{\partial t} = 0$ and the entropy inequality, i.e., $\frac{\partial S}{\partial t} \geq 0$ in an isolated system.

In order to properly generalize the dynamics to discrete domains like graphs, one requires to identify the state variable, an inner product structure as well as energy and entropy. Rather than only considering node features as the state variable, Gruber et al. (2023) extend the framework to higher-order clique cochains, including edges and cycles. Formally, let $\Omega_k$ be the set of $k$-cliques on a graph $\mathcal{G}$, which contains ordered, complete, subgraphs generated by $(k+1)$-nodes. For example, nodes, edges and triangles, correspond to the 0-clique, 1-clique and 2-clique respectively. The exterior derivative operator is denoted as $d_k : \mathfrak{F}(\Omega_k) \to \mathfrak{F}(\Omega_{k+1})$ where $\mathfrak{F}(\Omega)$ represents a function space over the domain $\Omega$. The specific structure of $\mathfrak{F}$ depends on the chosen inner product $\langle \cdot, \cdot \rangle$. The dual derivative $d_k^* : \mathfrak{F}(\Omega_{k+1}) \to \mathfrak{F}(\Omega_k)$ is given as the adjoint of the $d_k$ that satisfies $\langle d_k f, G \rangle_{k+1} = \langle f, d_k^* G \rangle_k$ for any $f \in \mathfrak{F}(\Omega_k), G \in \mathfrak{F}(\Omega_{k+1})$. One common choice of $\mathfrak{F}$ is the $L^2$ space, where one can derive $d_k^* = d_k^\top$. For example, $d_0$, is the graph gradient on nodes and $d_0^*$ becomes the graph divergence on edges. The classic graph Laplacian can be computed as $d_0^* d_0 = d_0^\top d_0 = \mathbf{G}^\top \mathbf{G}$ as we have shown in Section 2.

Instead, the paper pursues an inner product parameterized by positive definite matrices $\mathbf{A}_0, \mathbf{A}_1, ..., \mathbf{A}_k$ up to $k$-cliques. For example on node space $\Omega_0$, where the $L^2$ space has inner product $\mathbf{f}^\top \mathbf{g}$ for $\mathbf{f}, \mathbf{g} \in \mathfrak{F}(\Omega_0)$, the generalized inner product is given by $\mathbf{f}^\top \mathbf{A}_0 \mathbf{g}$. Under such choice, one can show $d_k^* = \mathbf{A}_k^{-1} d_k^\top \mathbf{A}_{k+1}$.

The state variable is set to be a node-edge feature pairs, denoted by $\mathbf{x} = (\mathbf{q}, \mathbf{p})$, which can be treated as the position and momentum of a phase space. Further, the following operators are utilized to extend the dynamics in (3) to graphs,

$$\tilde{\mathbf{L}} = \begin{pmatrix} 0 & -d_0^* \\ d_0 & 0 \end{pmatrix}, \qquad \tilde{\mathbf{G}} = \begin{pmatrix} d_0^* d_0 & 0 \\ 0 & d_1^* d_1 + d_0 d_0^* \end{pmatrix}, \quad \text{and} \quad \tilde{\mathbf{M}} = \begin{pmatrix} 0 & 0 \\ 0 & \mathbf{A}_1 d_1^* d_1 \mathbf{A}_1 \end{pmatrix}.$$

It can be verified that $\tilde{\mathbf{L}}$ is skew-symmetric and $\tilde{\mathbf{G}}, \tilde{\mathbf{M}}$ are symmetric positive definite with respect to the block-diagonal inner product parameterized by $\mathbf{A} = \text{diag}(\mathbf{A}_0, \mathbf{A}_1)$. Furthermore, let $\mathbf{X} = (\mathbf{Q}, \mathbf{P})$ denote the tuple of node and edge feature matrices, the energy considered is the total kinetic energy on both node and edge spaces, i.e., $E(\mathbf{X}) = \frac{1}{2}(\|\mathbf{Q}\|^2 + \|\mathbf{P}\|^2)$. The gradient with respect to the generalized inner product (called $\mathbf{A}$-gradient) can be computed as $\nabla_{\mathbf{A}} E(\mathbf{X}) = [\mathbf{A}_0^{-1}\frac{\partial E}{\partial \mathbf{Q}}, \mathbf{A}_1^{-1}\frac{\partial E}{\partial \mathbf{P}}]^\top = [\mathbf{A}_0^{-1}\mathbf{Q}, \mathbf{A}_1^{-1}\mathbf{P}]^\top$. For the Metriplectic dynamics, it is in general non-trivial to identify an entropy such that the degeneracy conditions hold. Hence the paper constructs a separate energy and entropy function pair as $E_m(\mathbf{X}) = f_E(\mathbf{Q}) + g_E(d_0 d_0^\top \mathbf{P})$ and $S_m(\mathbf{X}) = g_S(d_1^\top d_1 \mathbf{P})$, for some node function $f_E$ and edge functions $g_E, g_S$ applied channel-wise. The $\mathbf{A}$-gradient is derived as

$$\nabla_{\mathbf{A}} E_m(\mathbf{X}) = \begin{pmatrix} \mathbf{A}_0^{-1}\mathbf{1} \otimes \nabla_{\mathbf{A}} f_E(\mathbf{Q}) \\ d_0 d_0^\top \mathbf{1} \otimes \nabla_{\mathbf{A}} g_E(d_0 d_0^\top \mathbf{P}) \end{pmatrix}, \qquad \nabla_{\mathbf{A}} S_m(\mathbf{X}) = \begin{pmatrix} 0 \\ \mathbf{A}_1^{-1} d_1^\top d_1 \mathbf{1} \otimes \nabla_{\mathbf{A}} g_S(d_1 d_1^\top \mathbf{P}) \end{pmatrix}.$$

Importantly, the degeneracy conditions $\tilde{\mathbf{L}}\nabla_{\mathbf{A}} S = \tilde{\mathbf{M}}\nabla_{\mathbf{A}} E = 0$ are satisfied by construction.

Finally the generalized dynamics from (3) to graphs are

$$\text{Hamiltonian}_G : \quad \frac{\partial \mathbf{X}}{\partial t} = \tilde{\mathbf{L}}(\mathbf{X})\nabla_{\mathbf{A}} E(\mathbf{X});$$

$$\text{Gradient}_G : \quad \frac{\partial \mathbf{X}}{\partial t} = -\tilde{\mathbf{G}}(\mathbf{X})\nabla_{\mathbf{A}} E(\mathbf{X});$$

$$\text{Double bracket}_G : \quad \frac{\partial \mathbf{X}}{\partial t} = \tilde{\mathbf{L}}(\mathbf{X})\nabla_{\mathbf{A}} E(\mathbf{X}) + \tilde{\mathbf{L}}^2(\mathbf{X})\nabla_{\mathbf{A}} E(\mathbf{X});$$

$$\text{Metrplectic}_G : \quad \frac{\partial \mathbf{X}}{\partial t} = \tilde{\mathbf{L}}(\mathbf{X})\nabla_{\mathbf{A}} E_m(\mathbf{X}) + \tilde{\mathbf{M}}(\mathbf{X})\nabla_{\mathbf{A}} S_m(\mathbf{X}),$$

where the operators $\tilde{\mathbf{L}}(\mathbf{X}), \tilde{\mathbf{G}}(\mathbf{X}), \tilde{\mathbf{M}}(\mathbf{X})$ are state-dependent through attention mechanism to construct the metric tensor $\mathbf{A}_0, \mathbf{A}_1$, which thus parameterizes the exterior derivative operators.

**Remark 1** (Connection to GCN and GAT/GRAND). GCN can be seen as the $\text{Gradient}_G$ dynamics with $\mathbf{A}_0, \mathbf{A}_1$ parameterized by node degrees. Setting $\mathbf{A}_0$ as a diagonal matrix (on nodes) with diagonal entries $a_{0,ii} = \sqrt{\deg_i}$ and $\mathbf{A}_1$ as the identity matrix (over edges), we can verify $(d_0^* d_0 \mathbf{A}_0^{-1} \mathbf{Q})_i = \sum_{j \in \mathcal{N}_i} (\mathbf{q}_i/\deg_i - \mathbf{q}_j/\sqrt{\deg_i \deg_j}) = (\hat{\mathbf{L}} \mathbf{Q})_i$. In this case, $\text{Gradient}_G$ recovers the heat equation (with normalized Laplacian) when $\mathbf{P} = 0$ and thus leads to GCN under discretization.

Similarly, GAT can be seen as the same $\text{Gradient}_G$ dynamics while learning a metric structure $\mathbf{A}_0, \mathbf{A}_1$. That is, choose $a_{0,ii} = \sqrt{\sum_{j \in \mathcal{N}_i} \exp(\text{attn}(\mathbf{q}_i, \mathbf{q}_j))}$ and $a_{1,e_{ij}} = \exp(\text{attn}(\mathbf{q}_i, \mathbf{q}_j))$. Let the attention coefficient be $a(\mathbf{q}_i, \mathbf{q}_j) = a_{1,e_{ij}}/a_{0,ii}$. Then one can show $\text{Gradient}_G$ dynamics with $\mathbf{P} = 0$ corresponds to a (symmetrically) normalized version of GRAND.

This result demonstrates the irreversible nature of GCN or GAT/GRAND dynamics where energy dissipates, while other dynamics are either conservative or partially dissipative.

**Hamiltonian mechanics.** The *Hamiltonian mechanics* has also been considered in (Kang et al., 2023) where instead of using the edge features as the momentum, the work parameterizes the momentum with a neural network from the node features. This leads to distinction compared to the previous works in the evolution of the node features, which is decoupled from the graph structure. Let $\mathbf{Q}$ denote the node features and the momentum is computed as $\mathbf{P} = \text{MLP}_\Theta(\mathbf{Q})$. The Hamiltonian mechanics is determined by a Hamiltonian function $\mathcal{H}(\mathbf{Q}, \mathbf{P})$, which characterizes the total energy of the system. The dynamics is governed by the Hamiltonian equation

$$\frac{\partial \mathbf{Q}}{\partial t} = \frac{\partial \mathcal{H}}{\partial \mathbf{P}}, \qquad \frac{\partial \mathbf{P}}{\partial t} = -\frac{\partial \mathcal{H}}{\partial \mathbf{Q}}.$$

The paper motivates a variety of parameterization for the Hamiltonian $\mathcal{H}$. One specific choice of Hamiltonian is $\mathcal{H}(\mathbf{Q}, \mathbf{P}) = \text{tr}(\mathbf{P}^\top \mathbf{M}(\mathbf{Q}) \mathbf{P})$ where $\mathbf{M}(\mathbf{Q})$ represents the inverse metric tensor at $\mathbf{Q}$ (also learnable in the local neighbourhood). Its solution $\mathbf{Q}(t)$ recovers the geodesic (a locally shortest curve) on the manifold with metric parameterized by $\mathbf{M}(\mathbf{Q})^{-1}$ at $\mathbf{Q}$. Hence, nodes are effectively embedded to a (implicit) manifold space where the metric is learnable. Unlike previous methods, where the dynamics of the features depend on a coupling function regulated by the graph topology, here the evolution of $\mathbf{Q}$ is independent across the nodes. To further incorporate the graph structure, message passing by neighbourhood aggregation is performed after the feature evolution via Hamiltonian equation. Multiple layers of Hamiltonian dynamics and message passing are stacked to model complex geometries and node embeddings.

A follow-up work (Zhao et al., 2023a) extends the formulation by considering general graph-coupled Hamiltonian function $\mathcal{H}_G(\mathbf{Q}, \mathbf{P})$. For example, the paper considers a Hamiltonian function defined as the norm of the output from a two-layer GCN, where $\frac{\partial \mathcal{H}_G}{\partial \mathbf{Q}}, \frac{\partial \mathcal{H}_G}{\partial \mathbf{P}}$ are computed from auto-differentiation. Further, the paper studies various notions of stability on graphs from the theory of dynamical system, including BIBO, Lyapunov, structural and conservative stability. The work conducts a systematic analysis and comparison of proposed Hamiltonian-based dynamics with existing graph neural dynamics, such as GRAND, BLEND, Mean Curvature and Beltrami. It is found that the conservative Hamiltonian dynamics has shown improved robustness against adversarial attacks.

### 3.6 Dynamics as gradient flow

Most of the aforementioned designs of GNNs are inspired by evolution of some underlying dynamics. The learnable parameters, such as channel mixing, are usually added after discretization to increase the expressive power. In (Giovanni et al., 2023), the dynamics is instead given as the *gradient flow* of some learnable energy. The framework is general and includes many of the existing works as special cases (as long as the channel mixing matrix is symmetric)[†]. The parameterized energy takes the following form:

$$\mathcal{E}_\theta(\mathbf{X}) = \frac{1}{2}\text{tr}(\mathbf{X}^\top \mathbf{X} \mathbf{\Omega}) - \frac{1}{2}\text{tr}(\mathbf{X}^\top \mathbf{A} \mathbf{X} \mathbf{W}) + \varphi^0(\mathbf{X}, \mathbf{X}^0),$$

---

[†]Although many existing works can be written as gradient flow of some energy, their motivation comes mostly from the dynamics, not from the energy.

where $\mathbf{A}$ is the (normalized) adjacency matrix and $\mathbf{\Omega}, \mathbf{W} \in \mathbb{R}^{c \times c}$ are assumed to be symmetric[‡]. The first term determines the external forces exerted upon the system and the second term reflects the pairwise interactions while the last term quantifies the energy preserved by the source term $\mathbf{X}^0$. Although $\varphi^0$ can be general, the paper considers a form $\varphi^0(\mathbf{X}, \mathbf{X}^0) = \text{tr}(\mathbf{X}^\top \mathbf{X}^0 \tilde{\mathbf{W}})$. The gradient flow of $\mathcal{E}_\theta(\mathbf{X})$ yields the dynamics of the following general form,

$$\text{GRAFF}: \quad \frac{\partial \mathbf{X}}{\partial t} = -\nabla \mathcal{E}_\theta(\mathbf{X}) = -\mathbf{X}\mathbf{\Omega} + \mathbf{A}\mathbf{X}\mathbf{W} - \mathbf{X}^0 \tilde{\mathbf{W}}.$$

This formulation includes many of the existing dynamics-motivated GNNs. When $\mathbf{\Omega} = \mathbf{W}, \tilde{\mathbf{W}} = 0$, this corresponds to the evolution of (residual) GCN or GAT/GRAND (Chamberlain et al., 2021a) if $\mathbf{A}$ is constructed by attention mechanism. If $\tilde{\mathbf{W}} \neq 0$, this corresponds to the GRAND++ (Thorpe et al., 2022) and thus also the CGNN (Xhonneux et al., 2020). The decrease of the general energy does not necessarily lead to a decrease in the Dirichlet energy (which is a special case of $\mathcal{E}_\theta(\mathbf{X})$ with $\mathbf{\Omega} = \mathbf{W} = \mathbf{I}, \varphi^0 = 0$). This is mainly due to the occurrence of both attractive and repulsive effects along the positive and negative eigen-directions of $\mathbf{W}$. More formally, one decompose $\mathbf{W} = \mathbf{\Theta}_+^\top \mathbf{\Theta}_+ - \mathbf{\Theta}_-^\top \mathbf{\Theta}_-$ and rewrite the energy (without $\varphi^0$) as

$$\mathcal{E}_\theta(\mathbf{X}) = \frac{1}{2} \sum_{i \in \mathcal{V}} \langle \mathbf{x}_i, (\mathbf{\Omega} - \mathbf{W})\mathbf{x}_i \rangle + \frac{1}{4} \sum_{(i,j) \in \mathcal{E}} \|\mathbf{\Theta}_+ (\nabla \mathbf{X})_{i,j}\|^2 - \frac{1}{2} \sum_{(i,j) \in \mathcal{E}} \|\mathbf{\Theta}_- (\nabla \mathbf{F})_{i,j}\|^2.$$

It has been shown that the gradient flow minimizes the gradient along the positive eigen-directions (which leads to smoothing effect) while maximizing the gradient along the negative eigen-directions (which leads to sharpening effect). In contrast, minimizing the Dirichlet energy always induces smoothing effect because $\mathbf{W} = \mathbf{I}$ with only positive eigen-directions. This allows GRAFF to avoid oversmoothing and produce sharpening effects as long as the $\mathbf{W}$ has sufficiently large negative spectrum.

### 3.7 Multi-scale diffusion

Previous sections have mostly focused on dynamics with local diffusion. In other words, the signal/density at certain node changes depending on its immediate neighbourhood. The communication with distant nodes only happens when diffusion time is sufficiently large. Section 3.3 discusses dynamics that leverages non-local diffusion, but nonetheless mostly restricted to a single scale at each diffusion step, e.g., single $\alpha$ in fractional diffusion (Maskey et al., 2023) and $t_\theta$ for time derivative diffusion (Behmanesh et al., 2023). Multi-scale graph neural networks, such as ChebyNet (Defferrard et al., 2016), LanczosNet (Liao et al., 2018) and Framelet GNN (Zheng et al., 2021b) are capable of capturing multi-scale graph properties through spectral filtering on the graph Laplacian. The eigen-pairs of graph Laplacian encode structural information at different levels of granularity and separate processing of each resolution provides insights into both local and global patterns.

Several recent works have adapted the continuous dynamics formulation to multi-scale GNN. Han et al. (2022) introduce a multi-scale diffusion process via *graph framelet*. Apart from the multi-scale properties shared with other spectral GNNs, graph framelet further separates the low-pass from high-pass filters. Let $\mathcal{W}_{0,J} \in \mathbb{R}^{n \times n}$ denote the low-pass framelet transform and $\mathcal{W}_{r,j} \in \mathbb{R}^{n \times n}, r = 1, ..., R, j = 1, ..., J$ denote the high-pass framelet transforms, where $R$ is the number of high-pass filter banks and $J$ is the scale level. For a multi-channel graph signal $\mathbf{X} \in \mathbb{R}^{n \times c}$, $\mathcal{W}_{0,J}\mathbf{X}$ and $\mathcal{W}_{r,j}\mathbf{X}, r = 1, ..., R, j = 1, ..., J$ represent low-pass and high-pass framelet coefficients. For notation clarity let $\mathcal{I} = \{(0, J)\} \cup \{(r,j)\}_{1 \leq r \leq R, 1 \leq j \leq J}$ be the framelet index set. Due to the reconstruction property of framelets, it satisfies that $\sum_{(r,j) \in \mathcal{I}} \mathcal{W}_{r,j}^\top \mathcal{W}_{r,j}\mathbf{X} = \mathbf{X}$.

The spatial framelet diffusion proposed in (Han et al., 2022) is given as

$$\text{GradFUFG}: \quad \frac{\partial \mathbf{X}}{\partial t} = -\sum_{(r,j) \in \mathcal{I}} \left( \mathcal{W}_{r,j}^\top \mathcal{W}_{r,j}\mathbf{X}\mathbf{\Omega}_{r,j} - \mathcal{W}_{r,j}^\top \hat{\mathbf{A}} \mathcal{W}_{r,j}\mathbf{X}\mathbf{W}_{r,j} \right),$$

where $\mathbf{\Omega}_{r,j}, \mathbf{W}_{r,j}$ are symmetric channel mixing matrices. When $\mathbf{\Omega}_{r,j} = \mathbf{W}_{r,j} = \mathbf{I}$, by the tightness of framelet transform, GradFUFG reduces to the heat equation as $\frac{\partial \mathbf{X}}{\partial t} = -\hat{\mathbf{L}}\mathbf{X}$. The spatial-based framelet diffusion

---

[‡]The symmetric assumption is not required except for the purpose of simplification for the gradient derivation and analysis.

can be seen as gradient flow of a generalized Dirichlet energy, which also includes the parameterized energy considered by GRAFF (Giovanni et al., 2023) as a special case. It can be seen that, due to the separate modelling of low-pass and high-pass as well as different scale levels, the dynamics can adapt to datasets of different homophily levels as well as has the potential of avoiding oversmoothing.

## 4  On oversmoothing, oversquashing, heterophily and robustness of GNN dynamics

As we have briefly mentioned, the main hurdles for successful designs of GNN include oversmoothing, oversquashing, graph heterophily and adversarial perturbations. The framework of continuous GNNs allows principled analysis and treatment for the issues identified. In fact, many of the previously introduced dynamics are motivated to overcome the limitations of existing GNN designs. Nonetheless, there exist possible trade-offs such as between oversmoothing and oversquashing (Giraldo et al., 2022; Shao et al., 2023), which makes it challenging for GNN dynamics to avoid both phenomenon at the same time. Conceptually, common strategies for mitigating oversquashing relies on graph rewiring (Topping et al., 2022) that increases propagation strength, which likely enhances the smoothing effects. Apart from the trade-off, new problems may emerge as a result of resolving the existing ones. One example is the training difficulty (from gradient vanishing or explosion) associated with complex and long-range dynamics.

This section provides a holistic overview on these undesired behaviours of GNNs through the lens of continuous dynamics, along with a discussion on how existing approaches address the issues.

**Oversmoothing.** Oversmoothing refers to a phenomenon that node signals become progressively similar as a result of message passing, and converge to a state independent of the input signals. Although there are many different characterizations for oversmoothing, most if not all rely on the (normalized) Dirichlet energy that measures node similarity. Recall from Section 2, the Dirichlet energy is defined as $\mathcal{E}_{\mathrm{dir}}(\mathbf{X}) = \frac{1}{2} \sum_{(i,j) \in \mathcal{E}} \|\mathbf{x}_i/\sqrt{\deg_i} - \mathbf{x}_j/\sqrt{\deg_j}\|^2 = \mathrm{tr}(\mathbf{X}^\top \widehat{\mathbf{L}} \mathbf{X})$, where $\widehat{\mathbf{L}}$ is the symmetrically normalized Laplacian. Because oversmoothing claims a loss of distinguishability across the nodes, a natural quantification of oversmoothing is $\mathcal{E}_{\mathrm{dir}}(\mathbf{X}(t)) \to 0$ as $t \to \infty$. From the definition of the Dirichlet energy, in the limiting state, nodes with same degree collapse into a single representation. A stronger notion of oversmoothing has been considered in (Rusch et al., 2022; 2023a) requiring an exponential decay of the Dirichlet energy for oversmoothing to occur, i.e., $\mathcal{E}_{\mathrm{dir}}(\mathbf{X}(t)) \leq C_1 e^{-C_2 t}, \forall t > 0$ and for some constants $C_1, C_2 > 0$. This is equivalent to claiming that the system has a node-wise constant, exponentially stable equilibrium state. In this work, we focus on a notion of oversmoothing called low-frequency dominance (LFD) put forward by Giovanni et al. (2023). A dynamics is said to be low-frequency dominant if $\mathcal{E}_{\mathrm{dir}}(\mathbf{X}(t)/\|\mathbf{X}(t)\|) \to 0$. It can be readily verifies that when $\mathcal{E}_{\mathrm{dir}}(\mathbf{X}(t)) \to 0$, then $\mathcal{E}_{\mathrm{dir}}(\mathbf{X}(t)/\|\mathbf{X}(t)\|) \to 0$, while the reverse argument is false (see Appendix B.2 of Giovanni et al. (2023) more discussions). The normalization by $\|\mathbf{X}(t)\|$ reveals the dependence of asymptotic behaviour to the dominant spectrum of the dynamics, where low-frequency is related to the smoothing effect.

In the sense of LFD, it can be shown that the dynamics relying on (anisotropic) diffusion would incur oversmoothing in the limit, such as the linear versions of GRAND (Chamberlain et al., 2021a), BLEND (Chamberlain et al., 2021b), and DIFFormer (Wu et al., 2023b). See for example (Giovanni et al., 2023, Theorem B.4) for a formal proof. Dynamics that involves (non-smooth) edge indicators, like Mean Curvature, Beltrami flows (Song et al., 2022), $p$-Laplacian (Fu et al., 2022) and DIGNN (Fu et al., 2023) also correspond to LFD dynamics although the convergence is slower compared to (smooth) diffusion dynamics. Such a statement has been formalized in (Fu et al., 2023).

We summarize existing remedies for oversmoothing as follows.

(1) *Source term*: One remedy that steers the dynamics away from the low-frequency dominance is to include a source term. For example, in $p$-Laplacian (Fu et al., 2022) and DIGNN (Fu et al., 2023), the source term is implicitly added in terms of input dependent regularization, while GRAND++ (Thorpe et al., 2022) and CGNN (Xhonneux et al., 2020) explicitly inject the source information to the dynamics. The presence of a source term ensures the Dirichlet energy is lower bounded above zero where the limiting state is non-constant.

(2) *Non-smoothing dynamics*: Choosing a non-smoothing dynamics can help avoid oversmoothing, such as an oscillatory process as in PDE-GCN$_\mathrm{H}$ (Eliasof et al., 2021) and GraphCON (Rusch et al., 2022). Oscillatory system usually conserves rather than dissipates energy. In particular, it has been proved that the constant equilibrium state of the dynamics is not exponentially stable because any perturbation remains due to the energy conservation. The reversible dynamics, such as Hamiltonian and metriplectic in (Gruber et al., 2023) is also non-smoothing by construction.

(3) *Diffusion modulators*: Imposing external forces to counteract or modulate the diffusion mechanism is beneficial for mitigating oversmoothing. Examples include the high-pass filters in GREAD (Choi et al., 2023) and GradFUFG (Han et al., 2022), repulsive diffusion coefficients in ACMP (Wang et al., 2023) and ODNet (Lv et al., 2023), negative spectrum in GRAFF (Giovanni et al., 2023), reverse heat kernel in MHKG (Shao et al., 2023), anti-symmetry in A-DGN (Gravina et al., 2023), diffusion gating in $\mathrm{G}^2$ (Rusch et al., 2023b) and DeepGRAND (Nguyen et al., 2023). In essence, oversmoothing can be avoided in the above dynamics by diminishing smoothing effects in the limit of evolution.

(4) *Higher-order geometries*: Bodnar et al. (2022) demonstrate the cause of oversmoothing from the perspective of choosing a trivial geometry. In contrast, by enriching the graph with a higher-order sheaf topology, the asymptotic behaviour of the diffusion process can be better controlled. Indeed, the sheaf Dirichlet energy can be increased with a suitable choice of non-symmetric sheaf structure on graphs, thus avoiding oversmoothing under this regime (Bodnar et al., 2022, Proposition 17).

**Heterophily.** Unlike homophilic graphs where neighbouring nodes tend to share similar features and labels, nodes in heterophilic graphs usually exhibit dissimilar patterns compared to the neighbours. GNN that is dominated by smoothing effect is generally less preferred for such graphs. Following (Giovanni et al., 2023), the ability of a dynamics to adapt for heterophilic graphs is measured in terms of high-frequency dominance (HFD). A dynamics is said to be high-frequency dominant if $\mathcal{E}_{\mathrm{dir}}(\mathbf{X}(t)/\|\mathbf{X}(t)\|) \to \rho_L$, where $\rho_L \leq 2$ is the largest eigenvalue of $\widehat{\mathbf{L}}$. Intuitively, HFD dynamics is dominated by a sharpening effect and eventually converge to the highest-frequency eigenvectors of the Laplacian where node information gets separated. In this work, we equate the ability of a dynamics handling heterophily with its ability to become dominated by the highest frequency. Although HFD dynamics is also undesired due to information loss related to the low frequencies, here we focus on the ability, meaning that a system should have the flexibility to converge to the eigenspace associated with the highest frequency, which allows separability of bipartite graphs. Under such notion, dynamics that is purely smoothing cannot be HFD. The incorporation of a source term and (positive) graph re-weighting alone are not sufficient to turn a smoothing dynamics into a sharpening one even though empirically these strategies boost performance on heterophilic datasets. This is because the driving mechanism of the resulting system is still diffusion.

Existing dynamics that provably accommodates separating effects can be classified as follows.

(1) *Sharpening induced forces*: One scheme for a system to be HFD is to explicitly introduce sharpening induced forces. Most of the diffusion modulators identified for tackling oversmoothing amounts to incorporate such anti-smoothing forces, including GREAD (Choi et al., 2023), GradFUFG (Han et al., 2022), ACMP (Wang et al., 2023), ODNet (Lv et al., 2023), GRAFF (Giovanni et al., 2023), and MHKG (Shao et al., 2023). Conceptually, the dynamics is dominated by the high frequency if the magnitude of the sharpening effect surpasses the smoothing effect.

(2) *Higher-order geometries*: Once the graph is equipped with a sheaf topology, Bodnar et al. (2022) have shown that choosing a higher-dimensional stalks and non-symmetric restriction maps allows sheaf diffusion to gain linear separability in the limit for heterophilic graphs.

We highlight that other schemes, such as oscillatory dynamics in GraphCON (Rusch et al., 2022) and convection/advection process in (Zhao et al., 2023b; Eliasof et al., 2023) also demonstrate empirical success under heterophilic settings. Nevertheless, theoretical guarantee for theses schemes in adapting to heterophilic graphs is currently unavailable.

**Oversquashing.** The phenomenon of oversquashing has firstly been empirically observed by Alon & Yahav (2021) and later formally studied by Topping et al. (2022); Di Giovanni et al. (2023). Such a phenomenon has been identified as a key factor hindering expressive power of GNNs (Di Giovanni et al., 2023). Oversquashing concerns a scenario where long-range communication between distant nodes matters for the downstream tasks whereas message passing squashes information due to the exponentially growing size of the receptive field. A closely related concept is under-reaching, which refers to shallow GNN not being able to fully explore long-range interactions (Barceló et al., 2020). A critical difference between under-reaching and oversquashing is that the former can often be alleviated by increasing the depth of a GNN while the latter can occur even with deep GNNs. One major cause of such a phenomenon is the local diffusion process according to the given graph structure, where messages are exchanged only within the neighbourhood. This can be formally characterized by the sensitivity across nodes, measured through the Jacobian. More formally, consider a (discretized) dynamics $\mathbf{X}^{\ell+1} = \mathcal{A}_\mathcal{G}(\mathbf{X}^\ell)$, where $\mathcal{A}_\mathcal{G}$ is a coupling function that aggregates information based on input graph $\mathcal{G}$. The sensitivity between nodes $i, j \in \mathcal{V}$ after $\ell$ iterations of updates can be computed as $\left\| \frac{\partial \mathbf{x}_i^\ell}{\partial \mathbf{x}_j^0} \right\|_2$, where a smaller value indicates lower sensitivity and thus higher degree of oversquashing. In the simple case of isotropic diffusion $\mathcal{A}_\mathcal{G}(\mathbf{X}^\ell) = \sigma(\mathbf{A}\mathbf{X}^\ell\mathbf{W})$, one can show $\left\| \frac{\partial \mathbf{x}_i^\ell}{\partial \mathbf{x}_j^0} \right\|_2 \leq c^\ell (\mathbf{A}^\ell)_{i,j}$, where $\mathbf{A}^\ell$ denotes the $\ell$-th matrix power of $\mathbf{A}$ and $c > 0$ is a Lipschitz constant that depends on the activation function and $\mathbf{W}$. It can be readily noticed that the graph structure encoded in $\mathbf{A}$ critically determines the severity of oversquashing.

Below are strategies employed in GNN system that avoids oversquashing.

(1) *Graph topology modification*: One natural strategy in addressing the oversquashing is to enhance communications by rewiring the graph structure. GRAND (Chamberlain et al., 2021a) and BLEND (Chamberlain et al., 2021b) dynamically modify the connectivity according to the updated node features, which amounts to changing the spatial discretization over time. DIFFormer (Wu et al., 2023b), on the other hand, adopts a fully connected graph and thus avoids oversquashing by allowing message passing between all pairs of nodes.

(2) *Non-local dynamics*: Alternatively, non-local dynamics constructs dense graph communication function and thus mitigates oversquashing by design. This can be achieved for example by taking the fractional power of message passing matrix as in FLODE (Maskey et al., 2023), leveraging the dense quantum diffusion kernel in QDC (Markovich, 2023), learning the timestep of heat kernel in TIDE (Behmanesh et al., 2023), and capturing the entire path of diffusion as in G2TN (Toth et al., 2022).

(3) *Multi-scale and spectral filtering*: Multi-scale diffusion in GradFUFG (Han et al., 2022) has the potential to mitigate oversquashing by simultaneously accounting for information at different frequencies. Further, properly controlling the magnitude of high-pass versus low-pass filtering coefficients as in MHKG (Shao et al., 2023) (in the HFD setting) can provably improve the upper bound of the node sensitivity and thus oversquashing is alleviated.

Despite the success of aforementioned techniques in addressing oversquashing, a potential downside is the increased complexity of propagating information at each update, which is often associated with the more dense message passing matrix. We also remark that some methods, including GIND (Chen et al., 2022) and A-DGN (Gravina et al., 2023) are able to capture long-range dependencies without suffering from oversmoothing. In particular, the driving mechanisms for such purpose are the implicit diffusion in GIND and anti-symmetric weight in A-DGN where the former can be viewed as an infinite-depth GNN and the latter allows to build deep GNNs with constant sensitivity to avoid oversmoothing.

**Stability, gradient vanishing and explosion.** Lastly, we highlight several other potential pitfalls of GNN dynamics, that are often less explored in the literature compared to the previous issues, such as robustness and stability, and gradient vanishing and exploding during training.

Stability and robustness against adversarial attacks is a critical factor in assessing the performance of GNNs. For continuous dynamics, Song et al. (2022) verify that the graph neural diffusion, like GRAND (Chamberlain et al., 2021a) and BLEND (Chamberlain et al., 2021b), are empirically more stable to graph topology

perturbation compared to other discrete variants, which follows from the derived theoretical results based on heat kernel. Song et al. (2022) also prove that, due to the row-stochastic normalization of the diffusivity matrix in graph neural diffusion, the stability against node perturbation is guaranteed. In (Gravina et al., 2023), the robustness to the change in initial conditions is explicitly enforced with the required anti-symmetry (due to the zero real parts of the Jacobian eigenvalues). In (Zhao et al., 2023a), the conservative stability offered by Hamiltonian dynamics has shown enhanced robustness to adversarial attacks. In addition, Wu et al. (2023a) improve the generalization ability of GNNs with respect to topological distribution shifts, via local diffusion and global attention.

Another plight of deep neural networks are vanishing and exploding gradients, which refers to the situation when gradient exponentially converges to zero or infinity as a result of increasing depth. Because classic graph neural networks are often designed to be shallow as in GCN (Kipf & Welling, 2017) and GAT (Veličković et al., 2018), in order to avoid oversmoothing, gradient vanishing or explosion have received less attention in the GNN community. However, in the continuous regimes, especially when depth increases, these problems can emerge and severely escalate the training difficulty. In (Rusch et al., 2022), it has been verified that GraphCON is able to mitigate the vanishing and exploding gradient problems because the gradient of GraphCON is upper bounded at most quadratically in depth and is shown to be independent of the depth in terms of decaying behaviour. Due to the anti-symmetric channel mixing in A-DGN (Gravina et al., 2023), the magnitude of the gradient stays constant during backpropation and hence avoids the problem.

## 5 On training graph neural dynamics

Most of the works based on continuous GNN dynamics consider the following architecture, which is firstly introduced in GRAND (Chamberlain et al., 2021a).

$$\mathbf{X}(0) = \text{Emb}(\mathbf{X}_{\text{in}}), \qquad \mathbf{X}(T) = \mathbf{X}(0) + \int_0^T \frac{\partial \mathbf{X}(t)}{\partial t} dt, \qquad \mathbf{Y} = \text{Out}(\mathbf{X}(T)), \tag{4}$$

where $\frac{\partial \mathbf{X}(t)}{\partial t} = F_{\mathcal{G}}(\mathbf{X}, \nabla \mathbf{X})$ are the graph coupled, parameterized, differential equation introduced in Section 3.[§] $\text{Emb}(\cdot), \text{Out}(\cdot)$ are respectively the input embedding and output decoding functions, which are usually learnable. The formulation in (4) provides a general framework for various learning tasks on graphs, from node-level to graph-level, from transductive to inductive (see more discussions in Section 7). Similar to classic GNNs, the training for graph neural dynamics consists of forward and backward propagation. The forward propagation solves the graph differential equation via numerical integrators, while backward propagation consists of solving an adjoint differential equation to compute the gradient.

**Forward propagation.** There exist a variety of numerical solvers for differential equations, once the continuous dynamics is formulated on graphs. In fact, with the discrete differential operators defined on graphs, the PDE reduces to an ODE, which can be solved with standard numerical integrators. One natural strategy for discretizing a continuous dynamics is through *finite differences*. Particularly, the forward Euler and leapfrog methods are commonly employed for first-order and second-order ODE respectively. For a first-order ODE given by $\frac{\partial \mathbf{X}}{\partial t} = F(\mathbf{X}, t)$, the forward Euler discretization leverages forward finite time difference, which gives the update $\mathbf{X}(t+1) = \mathbf{X}(t) + \tau F(\mathbf{X}(t), t)$ for some stepsize $\tau > 0$. The forward Euler is an explicit method in that $\mathbf{X}(t+1)$ depends on $\mathbf{X}(t)$ explicitly. In contrast, backward Euler method is implicit by discretization $\mathbf{X}(t+1) = \mathbf{X}(t) + \tau F(\mathbf{X}(t+1), t+1)$, which involves solving a (nonlinear) system of equations to obtain $\mathbf{X}(t+1)$.

For a second-order ODE $\frac{\partial^2 \mathbf{X}}{\partial t^2} = F(\mathbf{X}, t)$, the leapfrog method that leverages centered finite differences yields the update $\mathbf{X}(t+1) = 2\mathbf{X}(t) - \mathbf{X}(t-1) + \tau^2 F(\mathbf{X}(t), t)$, which has been considered for PDE-GCN$_H$ (Eliasof et al., 2021). The leapfrog method can be equivalently derived by rewriting the second-order ODE into a system of first-order ODEs (concerning both the position and velocity) and then use forward Euler method to alternatively update the two. In GraphCON (Rusch et al., 2022), a similar idea has been applied for the more complex second-order dynamics.

---

[§]For second-order dynamics like in PDE-GCN$_H$ and GraphCON, the same formulation applies if the second-order ODE is rewritten as a system of first-order ODEs.

Higher-order methods employ high-order approximations for solving both first- or second-order ODEs and can be either explicit or implicit. Common choices include Runge–Kutta, which is a class of single-step methods that requires evaluating $F$ at multiple extrapolated state. One popular variant of Runge-Kutta method is Runge-Kutta 4 that balances the accuracy and efficiency. It computes $\mathbf{X}(t+1) = \mathbf{X} + \frac{\tau}{6}(\mathbf{K}_1 + 2\mathbf{K}_2 + 2\mathbf{K}_3 + \mathbf{K}_4)$ where $\mathbf{k}_1 = F(\mathbf{X}(t), t), \mathbf{K}_2 = F(\mathbf{X}(t) + \tau\mathbf{K}_1/2, t + \tau/2), \mathbf{K}_3 = F(\mathbf{X}(t) + \tau\mathbf{K}_2/2, t + \tau/2), \mathbf{K}_4 = F(\mathbf{X}(t) + \tau\mathbf{K}_3, t + \tau)$. Multi-step methods, such as Adams-Bashford, store multiple previous steps in order to approximate high-order derivatives. More advanced solvers set the adaptive stepsize based on the error estimated at each iteration. Dormand–Prince is a Runge-Kutta method with step-size control and has been widely adopted as the default solver for ODEs (Dormand, 1996).

**Backward propagation** To compute the gradient, it is generally memory-consuming to differentiate directly through the numerical integrators used in the forward pass. Instead, Chen et al. (2018) considered the *adjoint sensitivity method* (Pontryagin, 2018), which requires to solve an adjoint differential equation, which is more memory-efficient and accurate by explicitly controlling the numerical error. Such a scheme allows to decouple the forward and backward pass where the numerical solvers can be chosen differently. This is unlike the classic GNNs, where the complexity of the backward pass highly depends on the forward pass.

The forward and backward propagation schemes can be implemented through `torchdiffeq` package (Chen, 2018). Choosing suitable integration schemes depends on the trade-off between efficiency, accuracy and stability. From the comparisons in (Chamberlain et al., 2021a;b), it is shown that explicit solvers, like forward Euler is more efficient, at the cost of being less accurate and stable regarding the choice of step size. On the other hand, implicit method requires less steps to reach a desired accuracy but requiring high computational cost per step. We refer to `https://github.com/twitter-research/graph-neural-pde` for typical implementation of continuous GNNs.

## 6 Complexity and scalability of graph neural dynamics

In this section, we compare the computational complexity of introduced graph neural dynamics in this work. We assume the implementation follows (4) and hence only compares the cost for evaluating $\frac{\partial \mathbf{X}}{\partial t}$. Here we denote $\mathbf{X} \in \mathbb{R}^{n \times c}$ where $n$ is the number of nodes and $c$ as the number of channels after embedding. We also let $d$ be the dimension of attention transformation matrix.

Most dynamics of *anisotropic diffusion*, such as GRAND (Chamberlain et al., 2021a), require a time complexity of $O(|\mathcal{E}|c)$ without attention or $O(|\mathcal{E}|cd)$. For BLEND, $c$ is further augmented by the dimension of positional embedding, i.e., $O(|\mathcal{E}|(c + c_{\text{pos}}))$. Although DIFFormer (Wu et al., 2023b) requires to compute attention over the fully connected graph, a linear attention scheme is employed to lower the complexity to $O(ncd + n^2c)$ instead of $O(n^2cd)$. For *oscillatory* processes, including PDE-GCN$_H$ (Eliasof et al., 2021) and GraphCON (Rusch et al., 2022), the complexity is on the same order as $O(|\mathcal{E}|cd)$ by rewriting the propagation into a system of first-order dynamics.

*Non-local dynamics* generally demands a higher complexity by propagation information on a denser graph, such as QDC (Markovich, 2023), with a complexity $O(|\mathcal{E}'|c)$ where $|\mathcal{E}'|$ is the edge set corresponding to a denser graph. For the spectral-based propagation, like FLODE (Maskey et al., 2023), TIDE (Behmanesh et al., 2023), the cost scales with $O(n^2c)$ (after pre-computing eigendecomposition at a cost of $O(n^3)$). For G2TN (Toth et al., 2022) that requires to capture a long path of information with higher-order tensors, the complexity depends on the order of tensor $m$ and the number of propagation step $L$, i.e., $O(|\mathcal{E}|m^2L + |\mathcal{E}|mc)$.

*For diffusion dynamics with external forces*, the complexity depends on whether the added forces dominate the computation of diffusion. For CDE (Zhao et al., 2023b), GREAD (Choi et al., 2023), ACMP (Wang et al., 2023), ODNet (Hegselmann & Krause, 2002) and G$^2$ (Rusch et al., 2023b), A-DGN (Gravina et al., 2023), the complexity is still dominated by the diffusion term $O(|\mathcal{E}|c)$ (or $O(|\mathcal{E}|cd)$). For ADR-GNN (Eliasof et al., 2023) that requires to compute the edge weights further increases the complexity to $O((n + |\mathcal{E}|)c^2)$. Lastly, MHKG (Shao et al., 2023) operates ove the spectral domain and hence costs $O(n^2c)$.

For *Geometry-underpinned dynamics*, diffusion based on sheaf structure (Bodnar et al., 2022) requires at least $O(|\mathcal{E}|fc)$ (using diagonal maps) where $f$ is the stalk dimension. The bracket dynamics (Gruber et al., 2023)

Table 3: Summary of experiments conducted for graph neural dynamics. *Homo* and *Hetero* refers to experiments for node classification with homophilic and heterophilic graphs. *Inductive* refers to experiments validating generalization to unseen graphs. *Scalability* refers to experiments on large-scale graph benchmarks. *OSM/OSQ* refers to experiments validating the oversmoothing and oversquashing issues. *STAB* refers to experiments showing robustness and stability against perturbations.

| | Homo | Hetero | Inductive | Scalability | Graph-level | OSM | OSQ | STAB | Others experiments |
|---|---|---|---|---|---|---|---|---|---|
| GRAND[1] | ✔ | | | ✔ | | ✔ | | | |
| BLEND[2] | ✔ | | | ✔ | | | | | |
| Mean Curvature[3] | ✔ | | | | | ✔ | | ✔ | |
| Beltrami[3] | | | | | | | | | |
| $p$-Laplacian[4] | ✔ | ✔ | ✔ | ✔ | | | | ✔ | |
| DIFFormer[5] | ✔ | | | ✔ | | ✔ | | | Image/text classification, Spatial-temporal prediction |
| DIGNN[6] | ✔ | ✔ | ✔ | | ✔ | ✔ | | | |
| GRAND++[7] | ✔ | | | ✔ | | ✔ | | | Varying label rate |
| PDE-GCN[8] | ✔ | ✔ | ✔ | | | ✔ | | | Dense shape correspondence |
| GIND[9] | ✔ | ✔ | ✔ | | ✔ | | ✔ | | |
| GraphCON[10] | ✔ | ✔ | ✔ | | ✔ | ✔ | | | |
| FLODE[11] | ✔ | ✔ | | | | ✔ | | | Directed graphs |
| QDC[12] | ✔ | ✔ | | | | | | | |
| TIDE[13] | ✔ | ✔ | ✔ | ✔ | ✔ | ✔ | ✔ | | Geometric graphs |
| G2TN[14] | ✔ | | ✔ | | ✔ | | | | |
| CDE[15] | ✔ | ✔ | | | | | | | |
| GREAD[16] | ✔ | ✔ | | | | ✔ | | | |
| ACMP[17] | ✔ | ✔ | | | | ✔ | | ✔ | |
| ODNet[18] | ✔ | ✔ | | | | ✔ | | | Hypergraph diffusion, Network simplification |
| ADR-GNN[19] | ✔ | ✔ | | ✔ | | ✔ | | | Spatial-temporal prediction |
| G$^2$ [20] | ✔ | ✔ | | ✔ | | ✔ | | | Node regression |
| MHKG[21] | ✔ | ✔ | | ✔ | | ✔ | | | |
| A-GCN[22] | ✔ | ✔ | | | ✔ | ✔ | ✔ | | Tree-NeighborsMatch |
| NSD[23] | ✔ | ✔ | | | | ✔ | | | |
| Hamiltonian$_G$, etc.[24] | ✔ | | | | | ✔ | | | Trajectory prediction |
| HamGNN[25] | ✔ | ✔ | | ✔ | | ✔ | | | Link prediction, Mixed-geometry graphs |
| HamGNN[26] | ✔ | | ✔ | ✔ | | | | ✔ | |
| GRAFF[27] | ✔ | ✔ | | | | ✔ | | | |

[1](Chamberlain et al., 2021a), [2](Chamberlain et al., 2021b), [3](Song et al., 2022), [4](Fu et al., 2022), [5](Wu et al., 2023b), [6](Fu et al., 2023), [7](Thorpe et al., 2022), [8](Eliasof et al., 2021), [9](Chen et al., 2022), [10](Rusch et al., 2022), [11](Maskey et al., 2023), [12](Markovich, 2023), [13](Behmanesh et al., 2023), [14](Toth et al., 2022), [15](Zhao et al., 2023b), [16](Choi et al., 2023), [17](Wang et al., 2023), [18](Lv et al., 2023), [19](Eliasof et al., 2023), [20](Rusch et al., 2023b), [21](Shao et al., 2023), [22](Gravina et al., 2023), [23](Bodnar et al., 2022), [24](Gruber et al., 2023), [25](Kang et al., 2023), [26](Zhao et al., 2023a), [27](Giovanni et al., 2023)

scales linearly with the $k$-cliques and hence if only nodes and edges are considered as in the classic GNNs, the complexity is $O(|\mathcal{E}|cd)$, matching the diffusion dynamics with attention. Finally, GRAFF (Giovanni et al., 2023) requires a complexity of $O(n^2c + |\mathcal{E}|c)$ and GradFUFG (Han et al., 2022) has a complexity of $O(n^2c)$ due to the framelet transform.

In summary, the complexity/scalability of graph neural dynamics is mainly determined by the design of graph differential equations, which matches the discrete counterparts. In fact, using Euler discretization for solving the differential equations reduces the continuous dynamics into the classic discrete versions, such as GCN and GAT.

# 7 Empirical benchmarks and evaluations of graph neural dynamics

This section summarizes empirical experimental procedures for evaluating the performance of different graph neural dynamics. Because graph neural dynamics can be viewed as generalization of GNNs via continuous formulation, classic benchmarks for evaluating discrete GNNs are still applicable for the continuous GNNs.

**Node-level classification**    The most widely considered task is node classification, which aims to classify test nodes in a graph under semi-supervised setting (Kipf & Welling, 2017). For this task, the output $\mathbf{Y}$ in (4) is passed through a linear layer for predicting the probabilities for each class. Common benchmark datasets include citation networks, Cora (McCallum et al., 2000), Citeseer (Sen et al., 2008), Pubmed (Namata et al., 2012), co-authorship graphs, including CoauthorCS, CoauthorPhysics (Shchur et al., 2018), co-purchase graphs, including Computer, and Photo (McAuley et al., 2015). These graph datasets are known to be homophilic in the sense that neighbours tend to share similar features and labels. To verify whether a dynamics can adapt to heterophilic graphs, benchmarks including web-page link graphs, Cornell, Texas, Wisconsin (Pei et al., 2019), actor co-occurrence network (Tang et al., 2009), Wikipedia networks, including Chameleon and Squirrel (Rozemberczki et al., 2021). Previous datasets are only used for transductive learning, i.e., classifying nodes in the same graph that is used for training. For inductive learning setup, models are trained only on training graphs and are expected to predict node labels for test graphs. One popular dataset is PPI (Hamilton et al., 2017), where each graph corresponds to a different human tissue documenting protein-protein interactions.

**Graph-level classification**    Apart from the major experiments on node classification, some works consider graph-level prediction for evaluating the dynamics in extracting graph-level information. For this purpose, the output $\mathbf{Y}$ is passed through some pooling layer for aggregating the information across nodes for before final prediction. Commonly considered datasets, include MUTAG (Debnath et al., 1991), PTC (Helma et al., 2001), COX2 (Sutherland et al., 2003), NCI1 (Wale et al., 2008) which collect molecules or chemical compounds where each node represents an atom and edges represent chemical bonds; PROTEINS (Borgwardt et al., 2005), which collects proteins where each node represents amino acids and edges are constructed based on the spatial proximity. The task is to predict the properties at molecule, compound or protein level.

**Evaluation for oversmoothing and oversquashing.**    To evaluate whether a dynamics can potentially mitigate or even avoid *oversmoothing*, it is common to show node classification performance as the number of layers (or equivalently the integration time) increases. Alternatively, showing the evolution of Dirichlet energy in terms of depth is equally indicative of oversmoothing. On the other hand, it is generally difficult to explicitly measure the level of *oversquashing*. Nevertheless, many synthetic experiments have been designed to test whether a graph dynamics is able to capture long range dependencies. For example in (Gu et al., 2020; Chen et al., 2022; Di Giovanni et al., 2023), data is created to test the ability of classifying nodes in a given distance away. It is expected that a dynamics that mitigates oversquashing will suffer less from the performance degradation as the distance increases. For real applications, datasets that are known to exhibit strong long-range dependencies are often used for implicitly quantifying the degree of oversquashing for GNNs, including PascalVOC-SP, COCO-SP, PCQM-Contact, Peptides-func and Peptides-struct (Dwivedi et al., 2022).

**Evaluation for scalability and stability.**    GNNs are known to suffers from poor scalability to large graphs because the message passing scheme requires to input the entire graph each step. Common large-scale graph benchmarks include OGB-arxiv, OGB-proteins and OGB-products (Hu et al., 2020) where the number of nodes ranges from 13k to 2500k. It thus becomes critical to report the runtime and memory consumption of graph neural dynamics on such large-scale datasets. In addition to scalability, stability and robustness against adversarial perturbation is another important spectrum of benchmarking different dynamics. In general, various adversarial attacks, such as adding/removing nodes, injecting noise to node features, modifying graph topology, can be implemented through Graph Robustness Benchmark (GRB) (Zheng et al., 2021a). A dynamics that is robust to such attacks is generally preferred.

**Other experiments.**    Apart from the aforementioned benchmark experiments, some works consider tailored experiments showing the exclusive benefits of the proposed dynamics. This includes experiments on classifying images and texts where graphs are constructed based on feature similarities (Wu et al., 2023b), predicting node labels for spatial temporal graphs (Wu et al., 2023b; Eliasof et al., 2023), and processing geometric graphs (Eliasof et al., 2021; Behmanesh et al., 2023). Some works have generalized the dynamics to other graph types, including directed graphs (Maskey et al., 2023) and hypergraphs (Lv et al., 2023) and thus experiments for such graph types are considered.

We have summarized in Table 3 all the experiments performed for each dynamics introduced in Section 3.

## 8 Open research questions

The recent success of graph neural dynamics has presented numerous opportunities for future explorations. This section summarizes several exciting research directions that remain open.

**Systematic empirical comparisons of graph neural dynamics.** From the summary in Section 3, many dynamics share similar characteristics, especially the ones within the same categories. Moreover, the components of different dynamics can be combined together to form new dynamics. For example, the oscillatory process can be coupled with external forces like reaction and convection, and the augmentation of positional encoding is universal for all dynamics. Conducting controlled experiments that systematically compare the variety of graph neural dynamics through ablation and combination is an important future work for better understanding the utility of different components. This would provide principled guidelines for designing tailored dynamics for specific settings.

**Interpretation of spectral GNNs from the perspective of continuous dynamics.** We have summarized the dynamics mostly under the paradigm of spatial GNNs due to the nice interpretation of message passing as a form of diffusion. Spectral GNNs on the other hand, usually transforms the graph signals into coefficients weighted by eigenvectors of graph Laplacian (a process known as graph Fourier transform). Then filters are designed and learned to alter the frequency components of graph signals for downstream tasks. Although some existing works like (Han et al., 2022) have made preliminary attempts in formulating spectral GNNs through continuous dynamics, it is generally unknown whether there exists a unified framework for designing continuous dynamics on the spectral domain. Such a framework could be beneficial in bridging the gap between spatial and spectral methods and could further motivate designs of dynamics that take advantage of both paradigms.

**Dynamics with high-order graph structures and spatial derivatives.** Most existing continuous GNNs are limited to evolution over nodes, except for (Gruber et al., 2023) that considers higher-order cliques. Meanwhile graphs often encode more intricate topology where higher-order substructures, such as paths, triangles and cycles are crucial for downstream tasks (Thiede et al., 2021). For example, aromatic rings are cycle structures commonly appearing in molecule, which determines various chemical properties, such as solubility and acidity. Thus designing a dynamics aware of such local substructures is beneficial. In addition, current dynamics often rely on a coupling function that involves only the first-order spatial derivative (i.e., the gradient). It is thus interesting to investigate how to properly define higher-order spatial derivatives (e.g., the Hessian) and incorporate into the dynamics formulation.

**Explore the potential of graph neural dynamics for other applications.** Existing studies proposing continuous GNNs mostly focus on the node-level prediction tasks while the continuous formalism presents a general framework for modelling not limited to the evolution of nodes, but also edges, communities and even the entire graph. It is thus rewarding to adapt graph neural dynamics for other types of applications, such as link and graph-level prediction, anomaly detection, time series forecasting. For example, Elhag et al. (2022) leverage both linear and anisotropic diffusion for molecule property prediction; Bhaskar et al. (2023) utilize graph heat and wave equation for graph topology recovery; Eliasof et al. (2023) demonstrate the promise of using advection process for spatial temporal graph learning.

**Expressivity of graph neural dynamics.** The unprecedented success of GNNs have propelled researchers to study their expressive power in distinguishing non-isomorphic graphs (Xu et al., 2019; Morris et al., 2019), counting subgraph structures (Chen et al., 2020), etc. While the continuous formalism present a framework for interpreting and designing GNNs, there have been few works that characterize the expressivity of graph neural dynamics, especially on graph and subgraph levels. There also lacks theoretical understanding on how the choice of numerical integrators affects the performance of the neural dynamics. In addition, an equally fruitful direction is to explore whether the theory of dynamical systems, such as energy conservation and reversibility can be leveraged to design theoretically more powerful graph neural networks.

**Continuous formulation for other graph types.**  Most of the continuous GNNs focus primarily on dynamics over static, undirected, homogeneous graphs. Nevertheless, other graph types have also witnessed wide applicability, including *signed* or *directed graphs* (where edges can be negative or directed) (Derr et al., 2018; Huang et al., 2019; Monti et al., 2018; Tong et al., 2020), *heterogeneous graphs* (where nodes and edges have multiple types) (Wang et al., 2019; Ji et al., 2021), *geometric graphs*  (where nodes or edges respect geometric constraints) (Bronstein et al., 2021), *spatial-temporal graphs* (where graph topology also evolves in time) (Jin et al., 2023b;a). The continuous formulation of GNNs for these more complex graph types could be beneficial for both enhancing the understanding and representation power of existing GNNs. However, such generalization requires nontrivial efforts. For example, the dynamics should preserve symmetries for geometric graphs and account for both evolution of graph topology and signals for spatial-temporal graphs.

## 9    Conclusion

In this work, we conduct a comprehensive review of recent developments on continuous dynamics informed GNNs, an increasingly growing field that marries the theory of dynamical system and differential equation with graph representation learning. In particular, we provide mathematical formulations for a diverse range of graph neural dynamics, and show how the common plights of GNNs can be alleviated through the continuous formulation. We highlight several open challenges and fruitful research directions that warrant further exploration. We hope this survey brings attention to the potential of the continuous formalism of GNNs, and severs as a starting point for future endeavors in harnessing classic theories of continuous dynamics for enhancing the explainability and expressivity of GNN designs.

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
