# OpenReview forum: "From Continuous Dynamics to Graph Neural Networks: Neural Diffusion and Beyond"
_TMLR — Accepted by TMLR_

### Review · Reviewer_1KdX · 2023-11-18

**Summary Of Contributions:**

This submission gives a survey of existing works that take a continuous-dynamics perspective on graph neural networks. The authors start by introducing continuous diffusion processes and graph convolutional networks, and discuss how GCNs can be viewed as discretizations of a particular continuous diffusion process. They then discuss a large number of existing works that extend this interpretation, most of which involve defining a more complex continuous dynamical system over graphs and then discretizing it to obtain a graph neural network method. They also briefly discuss numerical solution method, and a set of problems for GNN methods that some of the earlier methods address. Finally, they conclude with some open problems for continuous-dynamics GNNs.

**Audience:**

Yes

**Broader Impact Concerns:**

I don't have any broader impact concerns for this submission.

**Claims And Evidence:**

Yes

**Requested Changes:**

### Qualify somewhat overbroad claims about GNNs

The paper currently frames all message-passing GNNs as being based on continuous diffusion, and has a number of broad statements about GNNs and message-passing, e.g. that message passing is "intrinsically linked to a physical process known as heat diffusion" (in the abstract) and "Diffusion dynamics has been shown to underpin the design of message passing and graph convolutional networks" (Section 3). This seems a bit overly simplistic to me. I agree many existing GNN architectures are related to heat diffusion (as discussed by Chamberlain et al., 2021) but it seems like a stretch to say that the message-passing technique is fundamentally linked to diffusion dynamics in general. For instance, it seems like some of the architectures in the [Message Passing Neural Networks framework (Gilmer et al. 2017)](https://proceedings.mlr.press/v70/gilmer17a) are not directly linked to any physical diffusion process.

Although these other GNN architectures aren't the main focus of this survey, I think this work should avoid implying that every GNN is fundamentally based on diffusion ideas. I'd like to see this addressed before recommending acceptance.

### Add context and definitions for some terms

Section 3 seems to assume that the reader is already familiar with some terms that aren't directly defined. I think the survey would be more accessible if it acknowledged that readers may not know these terms, and either summarized their definitions or explicitly referred to an existing definition. (This isn't the case for all terms; there are other terms the paper does give a good definition or summary of.)

Specific examples I noticed:

- "Oversmoothing" is referenced on page 2 and throughout section 3, but not actually defined or cited until Section 5. (More generally, I think it would be useful to give a brief description of the four types of problem referred to in Table 1 before the main content of section 3, instead of waiting until section 5.)
- On page 3, it's not clear what "a flux" is or what it means for it to exist.
- Page 6 doesn't explain what a "mean-curvature flow" is.
- Page 7 says that p-Laplacian diffusion corresponds to a "p-Dirichlet energy" but doesn't explain what that is.
- The "Quantum diffusion kernel" has a few terms like this:
  - What is a "system without a potential"? What kind of potential are we talking about?
  - What does it mean to be "equally delocalized"?
  - I also didn't understand how quantum interference occurs or how it leads to non-local message passing.
- The hypo-elliptic diffusion section uses some undefined algebraic terms like "an algebra lifting" and the $\mathbf{u}^{\otimes m}$ notation without explaining what they mean.

Given the breadth of the different methods covered, it's likely difficult to fully explain all of the terms used, but I think that at least in these cases readers could benefit from more context.

### Other minor specific suggestions for improvement

- The second paragraph of the introduction makes a number of statements about GNN performance degradation without citing any examples or analyses of this degradation.
- On page 3, shouldn't it just be "Fick's law" and not "the Fick's law"?
- I was a bit confused by the Beltrami flow discussion "Anisotropic diffusion" section on page 6. Does the Song et al Beltrami flow build on the BLEND flow or are they independent uses of Beltrami flows?
- On page 14, I didn't follow the claim "graphs usually possess more intricate geometries and viewed as discrete realization of general topological spaces". What does this mean?
- On page 15: Should the "over the domain $\omega$" be a capital $\Omega$?
- On page 20: Typo, "converses" instead of "conserves"
- On page 22: "Such" should not be capitalized

**Strengths And Weaknesses:**

**Strengths:**

*S1:* The survey is informative and covers a large number of different approaches and perspectives inspired by continuous dynamics. (I'm not an expert in this subarea, however, and was not previously familiar with most of the papers discussed here, so it's possible I'm unaware of additional relevant work, and my assessment might not be very reliable.)

*S2:* The introduction is clear and gives useful context for the continuous-dynamics perspective. The methods are grouped based on their central ideas, and most of the specific methods discussed seem to be explained well.

**Weaknesses:**

*W1:* The paper makes some broad statements about GNNs and message passing being connected to continuous dynamics as a general. However, there are also many GNNs that don't seem to have a natural continuous dynamics interpretation. I think the paper could be a bit more careful about its claims here.

*W2:* Some parts of Section 3 were still quite difficult for me to follow and seem to expect the reader to have a substantial background in continuous dynamics. I noticed multiple terms that were used without definitions or context. It also wasn't clear to me what the significance of each work was or how the method actually solves the problems it was motivated by. (Someone more familiar with the area might not have this problem, though.)

---

> ### Author Response · Authors · 2024-01-05
> **Responses**
>
> *1. "Qualify somewhat overbroad claims about GNNs". "For instance, it seems like some of the architectures in the Message Passing Neural Networks framework (Gilmer et al. 2017) are not directly linked to any physical diffusion process."*
>
> **Reply:** Thank you for such comment. We agree that there are some GNNs cannot be interpreted from diffusion process, for example, the spectral GNNs like ChebGCN. Nevertheless, message passing GNNs (the example you mention) are indeed related to diffusion process, which we have added to the paper in Page 4. We have also made a claim on the scope of GNNs we consider. For your convenience, below are the added texts.
>
>
> "In the more general setup with anisotropic diffusion coefficients $D(\mathbf x_i, \mathbf x_j, t)$, the diffusion process relates to the general form of message passing neural network (MPNN) (Gilmer et al. 2017). To see this, taking the Euler discretization of the diffusion equation with stepsize $\tau$, we can rewrite the diffusion as
> \begin{align*}
>     \mathbf x_i^{\ell + 1} &= \mathbf x_i^\ell + \tau \sum_{j: (i,j) \in \mathcal E} D(\mathbf x_i^\ell, \mathbf x_j^\ell) ( \mathbf x_j^\ell - \mathbf x_i^\ell) = U^\ell \big( \mathbf x_i^\ell, \sum_{j:(i,j) \in \mathcal E} M^\ell(\mathbf x_i^\ell, \mathbf x_j^\ell) \big),
> \end{align*}
> where $M(\mathbf x_i, \mathbf x_j) = D(\mathbf x_i, \mathbf x_j) (\mathbf x_j - \mathbf x_i)$ represents the message passing between two nodes $i, j$ and $U(\mathbf x_i, \mathbf m_i) = \mathbf x_i + \tau \mathbf m_i$ represents the message aggregation within the neighbourhood of node $i$. The channel-mixing matrix and nonlinear activation can be added for both the message passing and aggregation steps. It can be verified that GCN is a special instance of message passing neural network.
>
> We finally remark that there exist GNNs that cannot be easily interpreted within the continuous diffusion framework, mainly the spectral GNNs, which we exclude from the discussion of this work."
>
>
>
>
> *2. "Add context and definitions for some terms"*
>
> **Reply:** Thank you for the suggestions. We have now added more contexts for the terms you mentioned.
>
>
>
> *3. "I was a bit confused by the Beltrami flow discussion 'Anisotropic diffusion' section on page 6. Does the Song et al Beltrami flow build on the BLEND flow or are they independent uses of Beltrami flows?"*
>
> **Reply:** Thank you for the question. Both the BLEND flow and Beltrami flow by Song et al are motivated by the physical Beltrami flow. The difference is Song et al explicitly factor out the edge indicator from the attention matrix while BLEND does not. We have now added an explanation at the end of page 6.
>
>
> *4. "On page 14, I didn't follow the claim 'graphs usually possess more intricate geometries and viewed as discrete realization of general topological spaces'. What does this mean?"*
>
> **Reply:** We meant that graphs can be viewed as discrete approximation to some complex, continuous geometries (like Riemannian manifolds, topological spaces), where nodes/edges correspond to the points/sets on such spaces.
>
>
>
> *5. Other minor comments and suggestions.*
>
> **Reply:** We have modified the typos according to your suggestions.

---

> > ### Comment · Reviewer_1KdX · 2024-01-08
> > **Discussion**
> >
> > Thank you for your response, and I appreciate the revisions to the paper.
> >
> > **1.**
> > > Nevertheless, message passing GNNs (the example you mention) are indeed related to diffusion process
> >
> > I agree that the anisotropic diffusion (and the GCN) can be seen as special instances of MPNNs. But I don't think the reverse is true; I don't think it's necessarily true that every MPNN can be written as an anisotropic diffusion process. My point was that there exist message-passing neural networks that are not diffusion processes (or, at least, that are not naturally expressed as diffusion processes), and that the paper could be improved by acknowledging that not every message-passing GNN has a corresponding continuous dynamics model.
> >
> > As a concrete example, consider the Gated Graph Neural Network (Li et al. 2016), which is expressed as a MPNN by Gilmer et al. (2017) in Section 2. The GG-NN includes discrete-time GRU update steps instead of accumulating using +, and also also assumes the graph has discrete edge types and uses them to control the message passing operations. As such, it doesn't match the form of anisotropic diffusion you've written. Another example is the Molecular Graph Convolutions architecture (Kearnes et al. 2016) which is also a MPNN in Section 2 of Gilmer et al. (2017). This architecture updates node states using a ReLU activation at each discrete timestep, and also includes edge representations which are updated with ReLU, which again doesn't seem to correspond to an anisotropic diffusion process. There are even more examples like this as summarized in the "Graph Networks" framework ([Battaglia et al. 2018](https://arxiv.org/abs/1806.01261)).
> >
> > I still think the paper could be improved by acknowledging this explicitly. This could be done by expanding on the sentence at the end of page 4 referring to spectral GNNs to also mention the existence of more exotic MPNN / graph network architectures that don't fit nicely into the diffusion formalism.
> >
> > (By the way, it might also be worth citing an example of a spectral GNN in that remark as well, perhaps ChebGCN, so that readers know which architecture you are referring to there.)
> >
> > **2 and 3.**
> >
> > Thanks for the clarifications, that makes sense.
> >
> > **4.**
> >
> > Thanks for clarifying. So by "possess more intricate geometries" you mean that the graph could be viewed as an approximation of a more intricate geometric space? I'd suggest rewording this to make that a bit clearer. Perhaps something like "In fact, graphs can often be viewed as discrete approximations of more general topological spaces such as Riemannian manifolds, which posess complex continuous geometries."

---

> > > ### Author Response · Authors · 2024-01-09
> > > **Responses**
> > >
> > > We appreciate the further suggestions to improve the paper.
> > >
> > > **1**.  Thank you for the references. We have now explicitly acknowledged such MPNNs and spectral GNNs are excluded from the discussion of this work in the revised manuscript.
> > >
> > > **4**. We have now clarified in the revised manuscript according to your suggestion.

---

> > > > ### Comment · Reviewer_1KdX · 2024-01-09
> > > > **Thanks for the revisions**
> > > >
> > > > Thank you for the revisions. I believe all of my concerns with the submission have been suitably addressed.

---

### Review · Reviewer_uQos · 2023-12-20

**Summary Of Contributions:**

This comprehensive survey overviews the recent studies considering modeling GNN using continuous dynamics, i.e. neural diffusion and related approaches. In particular, the survey provides an introduction to the use of continuous dynamics in the context of GNN, a taxonomy of approaches and how continuous dynamics can address existing challenges in GNN in regards to oversmoothing, oversquashing, and heterophily. The survey ends up with future directions of research in the domain coined graph neural dynamics.

**Audience:**

Yes

**Broader Impact Concerns:**

I see no concerns in regards to ethical implications.

**Claims And Evidence:**

Yes

**Requested Changes:**

(Optional, if possible) I suggest revising section 3 to within each methodological framework restructure each section and provide all the diffusion formulations within a given framework and then discuss their strengths and limitations explicitly highlighting the different formulations and contrasting them to each other within the different dynamics formulations where possible.

Provide a table explaining all notations used in the paper to help the reader.

I would suggest include in the manuscript illustrative examples throughout the exposition. This could be an example illustrating the effect using different diffusion specifications (highlighting key different methodologies in section 3), illustrating the merits and limitations of ODE solvers (section 4), Exemplifying oversmoothing, oversquashing and heterophily (section 5) and how these effects are addressed by adequate specifications as discussed. This could be accompanied by code and some illustrative synthetic reproducible examples that would be very helpful to the reader.

Minor comments:
The paper is in need of a bit of proof reading, i.e.

Page 4 top:
“which applies channel wise” – it is unclear what channel wise refers to. Isn’t the elements of D scalar valued and positive in general and not just channel wise?
“can be also derived” -> “can also be derived”

Page 4 middle:
“Many of them are directly adapted from the existing physical processes.” – what existing physical processes are here specifically referred to in particular in context of oscillations and geometric diffusion? Perhaps this sentence can be removed. Perhaps rephrase to:
“Many of them are directly adapted from the existing physical processes as we will discuss below.”

Page 5:
“Blend further (Chamberlain et al., 2021b)” -> “Blend (Chamberlain et al., 2021b) further”

Page 7 middle:
“Similar in (Fu et al., 2022)” -> “Similar to (Fu et al., 2022)”

Page 9 bottom:
“that models the the average” -> “that models the average”

Page 16 bottom:
“Hence, node is effectively” -> “Hence, nodes are effectively”/ “Hence, each node is effectively”

Page 18 middle:
“the dynamics can adapt to datasets” -> “that the dynamics can adapt to datasets”

Page 18 bottom:
“by rewriting the the second-order” ->“by rewriting the second-order”

Page 19 top:
“similar idea has” -> “a similar idea has”/ “similar ideas have”

Page 22 middle:
“is gradient vanishing and exploding” -> “are vanishing and exploding gradients”

**Strengths And Weaknesses:**

Strengths:
* Table 1 provides a nice and useful overview and well outlines the foundation for the structure of section 3 describing the listed methods in general in a nice and coherent manner.
* The survey appears very comprehensive in terms of methodologies covered.
* The survey is in general well-structured and written.


Weaknesses:
* Some parts of the survey reads somewhat mechanically as paper A did this, then paper B this, then paper C did this etc. which makes section 3 rather monotone in some places to read. Would it be possible to restructure parts of these sections and for instance provide all the diffusion formulations within a given framework in terms of their mathematical specifications and then discuss their strengths and limitations explicitly highlighting the different formulations and contrasting them to each other within the different dynamics formulations?

* I was a bit disappointed reading section 6 which is rather short and given the otherwise comprehensive survey I would have liked to see this section expanded and further elaborated. I.e., could there also be future directions of research for instance in scaling these methodologies to large systems, enhance their reliability of the learned parameters, issues of explainability in how these modeling approaches work, methodologies for systematically benchmarking procedures and investigating what is the most adequate parameterization of graph neural dynamics? Overall, I feel this section is very short where I would here expect a much more elaborate discussion and further pointers to important open and future research directions.

---

> ### Author Response · Authors · 2024-01-05
> **Responses**
>
> *1. Restructure and explicitly comparing the different formulations.*
>
> **Reply:** Thank you for the comments. We have now added a paragraph at the start of Section 3, explicitly summarizing the key components and formulations for each class of dynamics we introduced. We have also added a section (Section 6), comparing the computational complexity of each dynamics.
>
>
> *2. Expansion and elaboration for section 6.*
>
> **Reply:** Thank you for your suggestion, we have now expanded section 6 to include future works on systematic empirical comparisons of graph neural dynamics as well as continuous framework for spectral GNNs.
>
> *3. Include illustrative examples throughout the exposition.*
>
> **Reply:** Thank you for the suggestion. We would like to highlight the goal of this work is towards understanding the formulation of GNNs through the lens of continuous dynamics, where we provide a variety of formulations, including the motivation and summarize them into different categories. We agree that empirically comparing the dynamics would be beneficial for understanding the difference among the formulations. However, we \textbf{sincerely} believe this is currently out of the scope of this survey, and would be pursued as part of our future work. Nevertheless, we have added a section (Section 7) and Table 3 discussing the empirical benchmarks used for validating the utility of the dynamics in the proposed papers.
>
>
>
>
> *4. "Provide a table explaining all notations."*
>
> **Reply:** Thank you for your suggestion, we have now added Table 1 summarizing the commonly used notations in the work.
>
>
>
> *5. Page 4 top: “which applies channel wise” – it is unclear what channel wise refers to. Isn’t the elements of D scalar valued and positive in general and not just channel wise?*
>
> **Reply:** Here, by 'channel wise', we mean the scalar value of $D$ is multiplied across all the channels of $\mathbf x$, which is clear from the equation at the bottom of page 3. To avoid confusion, we have removed such a statement.
>
>
>
> *6. Other minor comments and suggestions.*
>
> **Reply:** We have corrected the typos and modify according to your suggestions.

---

> > ### Comment · Reviewer_uQos · 2024-01-05
> > **Thanks for the revisions**
> >
> > I have checked the revised version of the manuscript and thank the authors for improving the manuscript addressing my major concerns. I also accept and understand that some of my suggested changes were out of scope and left unaddressed.
> >
> > Please check some of the wordings of the revised manuscript, in particular the use of
> > "a dynamics" this I believe should either be "a dynamic" or "dynamics" throughout.

---

> > > ### Author Response · Authors · 2024-01-07
> > > **Response to further comments**
> > >
> > > We appreciate your suggestion. We will make sure to correct the grammar error relating to "dynamics" in our final revision.

---

### Review · Reviewer_GiRx · 2023-12-24

**Summary Of Contributions:**

The paper is a survey paper covering papers studying the evolution of features in various GNN architectures the lens of viewing the architectures as the discretization of  a specific set of continuous dynamics.

  It opens by introducing concepts from the analysis of continuous dynamics like Fick's law, mass conservation, divergence, Laplace operator etc. and then relate the layerwise feature dynamics of GCN to the Euler discretization of the graph-diffusion version of the heat diffusion kernel (using graph definitions of divergence and gradient) with parametrization and non-linearities added after the discretization as an opening example.

  It then summarizes around 30 recent GNN papers which can explicitly made reference or can be analyized within the framework of

$\left[ \frac{\delta X}{\delta t},\frac{\delta^2 X}{\delta t^2} \right]\coloneqq \mathcal{F}_{\mathcal{G}}\left(X,\nabla X\right)$

with

$\mathcal{F}_{\mathcal{G}}$

 being a spatial coupling function implied by parametrization of neural networks, $$\frac{\delta X}{\delta t}$$ being the feature evolution throughout the network process (with layers being implied discretizations) and $$\nabla X$$ being the graph gradient of features $$(\nabla X)_{ij}\coloneqq x_i -x_j$$ , which are taken to be functions belonging to Hilbert spaces $$L^2 \left(\mathcal{V}\right),L^2 \left(\mathcal{E}\right)$$ on vertices and nodes respectively.

  Over 14 pages, the authors state the dynamics of each discussed paper and summarize the main innovation, implication or consideration that lead to the dynamic, as well as discussing the impact of parametrization and the discretization that converts the dynamics into a layerwise implementable neural network architecture (if not using NODEs).

  The final 4 pages are then spent summarizing insights on Oversmoothing, Oversquashing, the expressive power of the dynamics in the face of heterophily and training stability, as well as discussing open research questions.

**Audience:**

Yes

**Broader Impact Concerns:**

Review paper, none that I can see.

**Claims And Evidence:**

Yes

**Requested Changes:**

1. Please try to add a section showing the empirical (benchmark) impact of the different dynamics, where possible. I.e., it is worth adding a table summarizing what (if any) empirical evaluation was done (standard benchmark, custom experiments etc), discussing which papers are comparible to which others and then discussing those empirical insights to another.E.g., GRAFF compares against Sheaf,GRAND,GCN and others on relatively standard benchmarks as well as their custom experiment, and discussing their results together with other papers would make it easier for practitioners to decide which analysis line of research is worth digging deeper for them.
2. Similarly, the computational and parameter impact of the different dynamics can be discussed in the same section, as well as that of the different backprop and training methods. I understand that this is partially a paper about understanding, but the practical impact and trainability of each method is worthwhile additional context.
3. The other major change I'd request is checking at least some of the checkboxes of a systematic review, i.e.
    - state the explicit sources of consideration, search strategy etc
     - state the inclusion or exclusion criteria
    - include a list of papers considered and excluded/included according to these criteria
    - any other systematic review checklist item which makes sense

Systematic reviews come from studies of intervention, and this paper focuses on theoretical insights of ML models, so one shouldn't go crazy on this. However, the 4 steps above and maybe more from the general systematic review field would be value-adds in my opinion, both for future work which seeks to extend it and to explain the authors perspective further (and serve as a jump off list for other researchers). If it can be automated, making all the excluded papers doi-links could be a boon for readers as well.

**Strengths And Weaknesses:**

Strenghts:

1. Overall, I found this to be a thorough survey for the stated goal and scope (GNN papers using a continuous dynamics lense). Am am not immediately aware of important missing papers
2. The authors do a very good job summarizing the framework and each papers contributions. As a study guide, a launch-off point for a researcher wanting to find a footing or for a hypothetical student I would want to familarize with the topic, I would probably hand recommend this work. As far as I can tell, the summaries are also *correct*.
3. I want to highlight the "it is worth highlighting" snippets and explanations of concepts like sheafs or brackets which make it possible to read the paper without being deeply familiar with concepts from continuous dynamics. In terms of exposition and flow, this paper does a  really good job that is worth stressing

Weaknesses:

1. It is difficult to motivate "why should I care" from the studied examples alone if not already familiar with the literature. This could be alleviated by discussing empirical results where available
2. This is a personal trigger and borders on nitpicking, but the authors use the phrase "systematic and comprehensive" review. While I agree on the comprehensive, "systematic" has a meaning in science and would mean something checking at least *some* of e.g. the https://prisma-statement.org/ checklist.
3. There are some *minor* english flaws (superflous the) but nothing that is more than a nitpick and can't be dealt with using a judciously applied LLM or a native lector (I can't downlplay this enough, but it's worth fixing for nicety)
4. Finally, it *IS* a review, there is nothing new here, but it is sufficiently high quality that I find it valuable on its own

---

> ### Author Response · Authors · 2024-01-05
> **Responses**
>
> *1. "Please try to add a section showing the empirical (benchmark) impact of the different dynamics."*
>
> **Reply:** Thank you for your suggestions. Now we have added a section (Section 7 in the revised text), discussing the empirical benchmarks and evaluation for graph neural dynamics.
> We have also added a table (Table 3) comparing the experiments performed in the existing literature. Because each work has used different sets of baselines and problem settings, it is generally difficult to make comments on the comparisons without conducting experiments in a controlled setup. We believe this is an important future work that should be pursued in the future, as we have discussed in Section 8.
>
>
>
>
> *2. "Computational and parameter impact of different dynamics. And training and backprop methods."*
>
> **Reply:** Thank you for the suggestions.  we first add a section (Section 6), comparing the computational complexity of different dynamics. For training and backprop methods, we have revised Section 5 by explicitly discussing the forward and backward pass of training graph neural dynamics.
>
>
>
> *3. "checkboxes of a systematic review"*
>
> **Reply:** To avoid causing confusion, we have removed the term 'systematic' used to refer to the survey.

---

### Decision · Action_Editor_CvRm · 2024-02-01

**Recommendation:** Accept as is

**Comment:**

All reviewers were of the opinion that for the scope of the paper, the survey was thorough. All reviewers indicated they would recommend the paper to newcomers in the field. Therefore the recommendation is to extend the survey certification for this paper.

**Audience:**

All reviewers agree that the as a survey paper it can be of interest to the TMLR community.

**Claims And Evidence:**

All reviews agree that this survey paper makes correct claims.

---

> ### Author Response · Authors · 2024-02-21
> **Camera ready submission uploaded**
>
> Dear action editors and reviewers,
>
> We appreciate the time and effort in reviewing our paper. We have now uploaded the camera ready version of our paper. We have modified the organization paragraph to reflect the revised manuscript during rebuttal, along with some minor typos.
>
> Best, Authors